# Extremely strong polarization of an active asteroid (3200) Phaethon

Takashi Ito [1], Masateru Ishiguro[2], Tomoko Arai[3], Masataka Imai [4], Tomohiko Sekiguchi [5], Yoonsoo P. Bach [2], Yuna G. Kwon [2], Masanori Kobayashi [3], Ryo Ishimaru[3], Hiroyuki Naito [6], Makoto Watanabe [7] & Kiyoshi Kuramoto[4]

The near-Earth asteroid (3200) Phaethon is the parent body of the Geminid meteor stream. Phaethon is also an active asteroid with a very blue spectrum. We conducted polarimetric observations of this asteroid over a wide range of solar phase angles $\alpha$ during its close approach to the Earth in autumn 2016. Our observation revealed that Phaethon exhibits extremely large linear polarization: $P = 50.0 \pm 1.1\%$ at $\alpha = 106.5°$, and its maximum is even larger. The strong polarization implies that Phaethon's geometric albedo is lower than the current estimate obtained through radiometric observation. This possibility stems from the potential uncertainty in Phaethon's absolute magnitude. An alternative possibility is that relatively large grains (~300 μm in diameter, presumably due to extensive heating near its perihelion) dominate this asteroid's surface. In addition, the asteroid's surface porosity, if it is substantially large, can also be an effective cause of this polarization.

[1] Center for Computational Astrophysics, National Astronomical Observatory of Japan, Osawa 2-21-1, Mitaka, Tokyo 181–8588, Japan. [2] Department of Physics and Astronomy, Seoul National University, 1 Gwanak, Seoul 08826, Republic of Korea. [3] Planetary Exploration Research Center, Chiba Institute of Technology, Tsudanuma 2-17-1, Narashino, Chiba 275–0016, Japan. [4] Department of Cosmosciences, Graduate School of Science, Hokkaido University Kita-10, Nishi-8, Kita-ku, Sapporo 060–0810, Japan. [5] Asahikawa Campus, Hokkaido University of Education, Hokumon-cho 9, Asahikawa, Hokkaido 070–8621, Japan. [6] Nayoro Observatory, Nisshin 157-1, Nayoro, Hokkaido 096–0066, Japan. [7] Department of Applied Physics, Okayama University of Science, 1-1 Ridai-cho, Kita-ku, Okayama 700–0005, Japan. These authors contributed equally: Takashi Ito, Masateru Ishiguro, Tomoko Arai. Correspondence and requests for materials should be addressed to T.I. (email: tito.geoph.s@95.alumni.u-tokyo.ac.jp)

# (3200) Phaethon

(3200) Phaethon is a well-studied near-Earth asteroid. Ever since its discovery by a survey using the Infrared Astronomical Satellite (IRAS) in 1983[1], this asteroid has exhibited several interesting characteristics of small solar system bodies. Phaethon is an Apollo-type near-Earth asteroid that has large inclination ($i \sim 22°$), large eccentricity ($e \sim 0.89$), and small perihelion distance ($q \sim 0.14\,\mathrm{au}$). This asteroid is also recognized as the parent body of the Geminid meteor stream due to the orbital similarities they share[2,3], but no cometary activities such as coma have been detected[4,5] unlike in the parent bodies of other meteor streams. On the other hand, this asteroid exhibits weak but certain dust ejections near its perihelion passages[6,7], which is the reason why it is now regarded as an active asteroid[8].

A feature that makes Phaethon intriguing is its very blue spectrum[9,10]. This asteroid's spectrum is categorized into B-type in the SMASS II (Bus) classification[11] and F-type in the Tholen classification[3], characterized by a negative slope over 0.5–0.8 μm without any diagnostic absorption bands in the visible to near-infrared wavelengths[12]. Phaethon's blue spectrum is also similar to that of the Pallas family asteroids, particularly in the near-infrared wavelengths[9,10]. This is one item of evidence of the connection between this asteroid and the Pallas family. It is suggested that thermally metamorphosed CI/CM chondrites[12] or CK4 chondrites[10,13] are the meteorite analogue of Phaethon due to their spectral similarity.

Another interesting aspect of Phaethon is that, this asteroid seems to possess at least one disruption fragment, (155140) 2005 UD, deduced from the two asteroids' strong orbital similarity and their surface color affinities[14–16]. Multicolor photometry of (155140) 2005 UD indicates that the surface color of this object is inhomogeneous[16]. The surface heterogeneity of this fragment may be related to the spectral variability recognized on Phaethon's surface[12], serving as evidence of the past breakup event that split them.

The curious surface property of this asteroid, together with the existence of a fragment, is an important clue for understanding the dynamical and thermal evolution of the near-Earth asteroids in this orbital category, and not just of Phaethon, whose dynamical lifetime is generally as short as some million years[17,18]. However, it is also true that many unknowns and uncertainties remain to be cleared up. For example, we still have little understanding of what microscopic structure or physical process yields such very blue spectra of Phaethon. The details of the mechanism that causes the sporadic dust ejections from Phaethon, and the mechanism that made the fragment split from the parent body, are not well understood either, although it is suggested that strong solar heating is involved[19].

We can solve several of the above-stated enigmas of Phaethon (and those of the B-type asteroids collectively) by investigating their surface polarimetric properties. Polarimetric studies of airless bodies are generally useful for understanding their surface physical properties, particularly geometric albedo and grain size. For example, we know that geometric albedo and polarization degree of the small solar system bodies have a strong correlation[20]. Also, the maximum values of the linear polarization degree ($P_{max}$) and the solar phase angle ($\alpha$) where $P_{max}$ happens are correlated with grain size[21,22]. While spectroscopic observation measures the reflected spectrum from the surface of the object, its result depends not only on the surface texture but also chemical composition of the surface material. In general it is not easy to decouple these combined effects just from spectroscopic observation. Therefore, polarimetric observation that directly measures the status of light scattering, which strongly depends on the surface texture, is complementary to and sometimes superior

to spectroscopic observation in terms of studies of the small solar system bodies. So far, most polarimetric studies of the small solar system bodies have been performed at small to moderate solar phase angles such as $\alpha < 35°$[23]. Polarimetric measurement of the small bodies at a large solar phase angle is technically difficult, not only because the observational opportunities are limited to some near-Earth asteroids that get inside Earth's orbits, but also because the observation should be conducted at small solar elongation angles. However, as polarimetric measurement of the small bodies over a wide range of $\alpha$ better reveals their surface material property, its implementation is always desirable whenever it is feasible.

In this paper, we report the result of our series of polarimetric observations of Phaethon over a wide range of solar phase angles. Our observation revealed that this asteroid exhibits a very strong linear polarization (>50%). This implies that Phaethon's geometric albedo is lower than the current estimate. An alternative is that relatively large grains (~300 μm in diameter) dominate this asteroid's surface.

## Results

**Dependence of polarization on solar phase angle**. We carried out our series of polarimetric observations using the 1.6-m Pirka telescope at the Nayoro Observatory in Hokkaido, Japan, over six nights from September to November 2016. We provide details of the observation and analysis procedure in Methods. As an initial result, we made a plot of Phaethon's polarization degree $P_r$ as a function of solar phase angle $\alpha$ (Fig. 1). For reference, we show measurement results in past studies of some other solar system objects in this figure: Mercury, a Q-type near-Earth asteroid (1566) Icarus, and several others that are renowned for their large $P_r$.

In general, linear polarization degree of the small solar system bodies has a local maximum ($P_{max}$) at large $\alpha$ and a local minimum ($P_{min}$) at small $\alpha$[24], although the values of $P_{max}$, $P_{min}$, and $\alpha$ where the extremums happen differ from object to object. The $P_r(\alpha)$ curve for Mercury in Fig. 1 typically exemplifies what we just described—we clearly recognize $P_{min}$ at $\alpha \sim 10°$ and $P_{max}$ at $\alpha \sim 90°$. For Icarus, although there is no observational point that tells us the location of $P_{min}$, we see $P_{max}$ at $\alpha \sim 120°$. We may read $P_{min}$ from the plots for Deimos (0.43 μm).

Compared with the other objects plotted on Fig. 1, we easily see how strong Phaethon's polarization degree is. Its $P_r$ exhibits a steep increase toward large $\alpha$. Here, we must be particularly aware that the observed largest value of $P_r$ (~50% at $\alpha = 106.5°$) is not equivalent to Phaethon's $P_{max}$. As is expected from Fig. 1, $P_{max}$ of this asteroid is probably located beyond our observational coverage, at much larger $\alpha$ than 106.5°. Consequently, we can say that $P_{max}$ of Phaethon is substantially larger than 50%.

**Albedo and maximum polarization degree**. As for small solar system bodies, $P_{max}$ has a correlation inverse to the geometric albedo $p_V$ in general: Material that has large $p_V$ tends to have small $P_{max}$. This is so-called Umow's law[25] which is caused by the general fact that multiple scattering of light is more effective on surfaces with high albedo than on surfaces with low albedo. This makes the polarization degree of the surface with higher albedo weaker, and makes that with lower albedo stronger[26]. Using Phaethon's currently estimated geometric albedo ($p_V = 0.122 \pm 0.008$[27]), we made another plot that shows the dependence of $P_{max}$ on albedo (Fig. 2). In this figure, we again gathered data from several bodies and materials in addition to Phaethon: Mercury, (1566) Icarus, comets 2P and 209P as in Fig. 1, and various terrestrial, meteoritic, and lunar samples obtained from laboratory observations[28,29]. We included 2P and 209P in this

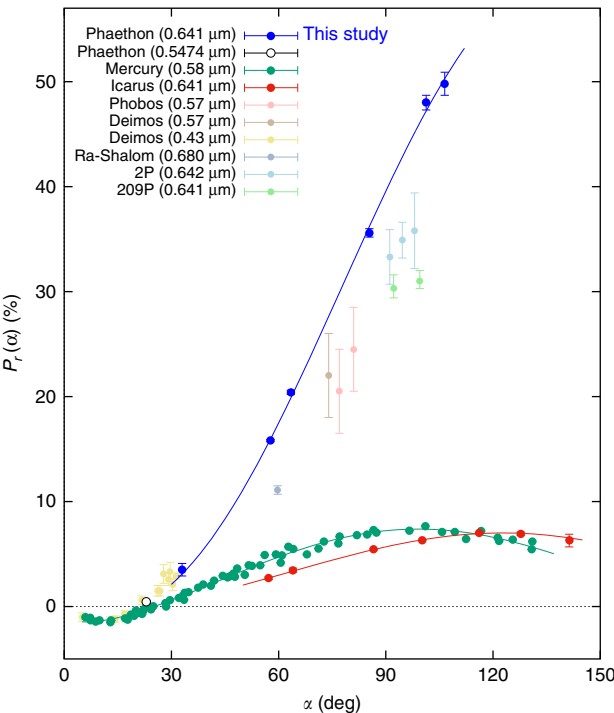

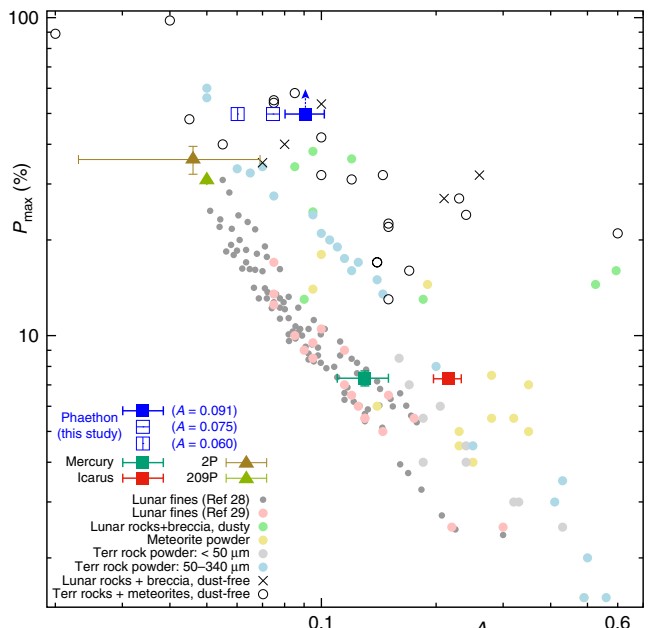

**Fig. 1** Dependence of the linear polarization degree $P_r$ of Phaethon (the blue-filled circles) on solar phase angle ($\alpha$) in the $R_C$-band centered at the wavelength of 0.641 μm. For comparison, $P_r$ values of some other solar system objects are also shown: Phaethon measured by other authors (0.5474 μm)[47], Mercury[33,62,63], (1566) Icarus[31], Phobos[64], Deimos (0.57 μm)[64], Deimos (0.43 μm)[65], (2100) Ra-Shalom[66], 2P/Encke in 0.642 μm ("observed")[67], and 209P/Linear (nucleus)[68]. Note that for Phaethon, Icarus, and Mercury, we fit the observational data by a regular, analytic function called the Lumme and Muinonen function[31,69]. Error bars of Phaethon's $P_r$ represent the sum of random errors and systematic errors that our polarimetric measurement contains. They are calculated in a manner described in Methods (see subsection Estimate of errors). Error bars of other objects' $P_r$ are adopted from the literature. The data for Phaethon are available from Table 1, and the data for Icarus are available from Supplementary Table 1. Dependence of Phaethon's linear polarization degree on its rotation phase is presented as Supplementary Figure 1, together with the actual data values tabulated in Supplementary Table 4

**Fig. 2** Relationship between $A$ (the geometric albedo measured at $\alpha = 5°$) and $P_{max}$ for Phaethon and other objects. In this figure, Phaethon is represented by three kinds of blue squares: The blue-filled square that uses the current albedo estimate in ref. [27] and its error bars, the blue open square with a horizontal bar that uses the albedo estimate brought by a recent radar observation[36], and the blue open square with a vertical bar that uses an albedo estimate calculated from the absolute magnitude value of $H = 14.6$ presented in ref. [38]. See Discussion for the latter two estimates. Note that the vertical value for Phaethon is not actually $P_{max}$, but the observed largest value of $P_r$ during our observation (Fig. 1). Therefore, we have added an upper arrow to the blue-filled square for showing that Phaethon's $P_{max}$ is larger than this value. Data for the terrestrial, meteoritic, and lunar samples are all taken from the tables described on ref. [29] except for those designated as "Lunar fines Ref [28]" adopted from ref. [28]'s Fig. 2. The lunar fine data adopted from ref. [29]'s Table 1 are designated as "Lunar fines Ref [29]." Note that in the legend we use the abbreviation "Terr rock" for terrestrial rock. The data for Mercury is from ref. [33], and that for (1566) Icarus is from ref. [31]. The albedo of 2P is adopted from ref. [70], and that of 209P is from ref. [68]. Note also that wavelengths for each of the measurements differ from sample to sample. Error bars seen on symbols of Icarus, Mercury, 209P, and 2P are adopted from the literature. The data values for Phaethon and Icarus plotted in this figure are available from Supplementary Table 2

figure because their $P_r$ values plotted in Fig. 1 can be regarded as being close to $P_{max}$. We excluded Phobos, Deimos, and (2100) Ra-Shalom because it is not certain whether their $P_r$ values in Fig. 1 are close to $P_{max}$.

Note that the albedo used in the horizontal axis of Fig. 2 is not $p_V$ itself, but the geometric albedo measured at $\alpha = 5°$. Hereafter, we call this $A$. This conversion is done to avoid the so-called opposition effect[30] that is extraordinarily eminent around $\alpha \sim 0°$. To convert Phaethon's $p_V$ into $A$, we adopted the ratio of reflectance intensity ($\mathscr{I}$) at $\alpha = 0.3°$ and $5°$ for three asteroids ((24) Themis, (47) Aglaja, and (59) Elpis) presented in ref. [30]. They are all B-type in the SMASS II classification, as is Phaethon. The arithmetic average of their intensity ratios $\left\langle \frac{\mathscr{I}(0.3°)}{\mathscr{I}(5°)} \right\rangle$ turns out to be 1.31. Using this value, we carried out the conversion of Phaethon's albedo from $p_V$ to $A$ as $A = \frac{p_V}{1.31} \sim 0.0931$. A similar conversion is applied to (1566) Icarus[31].

When drawing Fig. 2, we need to be aware that Phaethon's visual geometric albedo ($p_V = 0.122 \pm 0.008$) reported in ref. [27] is defined in the V-band (centered at 0.545 μm), not in the $R_C$-band (centered at 0.641 μm) where our observation was carried out. This wavelength difference can affect albedo. For deriving

Phaethon's geometric albedo in the $R_C$-band from the reported $p_V$ in the V-band, we adopted the averaged spectral intensity-difference of Phaethon between 0.641 μm and 0.545 μm measured in ref. [32]. This conversion yields an estimate of Phaethon's albedo in the $R_C$-band as $A = 0.0910$, which is slightly lower than that in the V-band ($A = 0.0931$). As for (1566) Icarus, we used the observed $P_{max}$ in the V-band and the $A$ value at this wavelength[31]. As for Mercury, we adopted the values described in ref. [33]: $A = 0.130$ at $\alpha = 5°$ in 0.585 μm. As for the lunar fine data presented in ref. [28]'s Fig. 2, there is no specific description of wavelength in the paper. Therefore, we assumed 0.6 μm ("orange light") depicted in a closely relevant paper by the same author[34] on the same subject. For the comets 2P and 209P, the measurement wavelengths are denoted in Fig. 1. All the laboratory samples described in ref. [29] are measured in the wavelength of 0.58 μm.

Figure 2 largely realizes Umow's law, representing the inverse correlation between geometric albedo and $P_{max}$. As you see in this

figure, the albedo of Mercury and Icarus is moderate, so is their $P_{max}$. The albedo of comets 2P and 209P is very low, and their $P_{max}$ is large. On the other hand, Phaethon exhibits very large $P_{max}$ while its albedo is not as low as that of comets 2P and 209P. This obviously looks odd, and it must be accounted for by reasonable physical explanation.

## Discussion

As we see in Figs. 1 and 2, our polarimetric measurement showed that Phaethon possesses a very strong linear polarization on its surface. The straightforward application of Umow's law to this result tells us that, Phaethon's geometric albedo can be lower than what is currently estimated. Phaethon's geometric albedo, as well as that of many other asteroids, is estimated through the combination of radiometric observation in infrared wavelengths and photometric observation in visible wavelengths. Accuracy of an asteroid's albedo estimate in this way largely depends on how accurately its absolute magnitude ($H$) is determined. And, accuracy of the absolute magnitude determination depends on the accuracy of phase curve function determined by photometric observation in visible wavelengths at solar phase angle $\alpha$ from small to large values. However, ground-based observation of Phaethon at very small solar phase angle is intrinsically difficult due to the relative orbital configuration between this asteroid and the Earth. Therefore, Phaethon's absolute magnitude determination is based on the phase curve observations whose minimum solar phase angle is no smaller than $12°$[27,35]. This means that the influence of the opposition effect that can happen at very small $\alpha$ has not been directly measured. Consequently, inclusion of uncertainty into Phaethon's absolute magnitude is inevitable. Hence, Phaethon's albedo estimate can contain a relatively large uncertainty as long as it comes through radiometric measurement.

Recently, Phaethon's effective diameter ($D$) was determined as $D = 5.7$ km through a radar observation at its close approach to the Earth in December 2017[36]. Measurement of size and shape of near-Earth asteroids through active radar observation is known to be reliable[37], and we presume the estimated effective diameter is accurate. In ref.[36], Phaethon's geometric albedo is preliminarily revised as 0.10 by assuming the absolute magnitude $H = 14.3$. On the other hand, several different estimates of Phaethon's absolute magnitude have been published. One of the faint estimates is $H = 14.6$[38]. If we apply the combination of $D = 5.7$ km and $H = 14.6$ to the common formula[39] between asteroid's diameter $D$, geometric albedo $p_V$, and absolute magnitude $H$

$$\log_{10} D = 0.5(6.259 - \log_{10} p_V - 0.4H), \quad (1)$$

we obtain $p_V = 0.081$ for Phaethon, a much smaller albedo than the currently accepted value. Using these albedo estimates, we placed two more symbols for Phaethon in Fig. 2.

As for the uncertainty of absolute magnitude determination based on phase curves, we know that it can reach 0.1 magnitude even for asteroids whose phase curve is accurately measured down to very small solar phase angle[30]. Recalling the fact that Phaethon's phase curve is measured only down to $\alpha \sim 12°$, it is not hard to imagine that its absolute magnitude estimate contains uncertainties substantially larger than 0.1 magnitude. This fact endorses the prospect of Phaethon's albedo uncertainty: This asteroid's geometric albedo is presumably lower than the currently accepted estimate. The lower albedo can cause Phaethon's strong $P_{max}$ that our polarimetric measurement found out.

If the current albedo estimate of Phaethon is accurate enough and not quite low, what else could cause its strong polarization? In this case we would direct our attention to the fact that in Fig. 2, the terrestrial samples with larger grains (50–340 μm) yield

stronger $P_{max}$ than those with smaller grains (<50 μm) and lunar fines. In other words, we would suspect that Phaethon's strong polarization has something to do with its surface grain size.

A few empirical formulas are known between the grain size $d$ of regolith-like material and the converted albedo $A$ obtained from various laboratory measurements. When expressing $d$ in μm, $P_{max}$ in %, and $A$ at the wavelength of 0.65 μm in %, one of the formulas[40] is expressed as

$$d = 0.03 \exp\left[2.9\left(\log_{10} A + 0.845 \log_{10} 10 P_{max}\right)\right]. \quad (2)$$

Equation (2) tells us that the larger the grain is, the stronger the polarization gets, as long as albedo remains constant. When larger grains dominate an object's surface, there would be fewer grains down to unit optical depth. Consequently, multiple scattering of incident light would happen less often, which leads to stronger polarization. Substituting the actual values of Phaethon ($A = 9.1\%$ derived from the current albedo estimate, and $P_{max} = 50\%$ from the observed largest $P_r$) into Eq. (2), we get $d \sim 360$ μm. Although uncertainty is unavoidable as to how appropriate it is to apply Eq. (2) obtained from laboratory measurements of terrestrial and lunar samples to the surface state of a small body such as Phaethon, it is worth noting that the estimated value ($d \sim 360$ μm) belongs to the largest category among the laboratory samples. Incidentally, let us note that the newly estimated albedo value (0.10) through the radar observation[36] yields $d \sim 280$ μm.

The hypothesis of the dominance of larger grains on Phaethon's surface has an affinity as well as an inconsistency with observational facts. As for the affinity, let us remember that Phaethon is famous for its very blue spectrum[9,10,12]. And we know from experiments that the spectra of meteoritic and asteroidal materials tend to get bluer when we increase their effective grain size[13,41,42]. A possible mechanism that produces the large grains is sintering. Phaethon's surface can be heated up to 1000 K during its perihelion passage[43], which is as high as the metamorphic temperature of some types of carbonaceous chondrites[44,45]. Such an extreme heating can cause sintering on this asteroid's surface, making the grain coarsening happen[46].

Let us mention the inconsistency. While our polarimetric measurement result obtained at large solar phase angle may imply the possible dominance of relatively large grains on Phaethon's surface, polarimetric measurements in the negative branch obtained at smaller phase angle presented in past studies[23,47] suggest that this asteroid behaves rather typically as B-type with a moderate inversion phase angle (which divides the negative and positive polarimetric branches). From this viewpoint, Phaethon's surface texture does not seem quite dominated by large grains. Currently, this discrepancy cannot be solved by our observation result alone, and it should be further investigated in future.

Let us add yet another possibility that can enhance Phaethon's polarization degree: Large surface porosity. It has been numerically confirmed that large porosity significantly increases polarization of material surface[48,49]. We also know that this trend holds true regardless of wavelength of incident light[50,51], although its theoretical understanding is not yet completely established, particularly when the wavelength is shorter than the characteristic size of light scatterers on the object surface. In terms of geometry, surface porosity can be larger in general when the surface grain shape is rough or irregular regardless of the grains' average size. Although the irregularity of surface grains itself may not significantly affect the $P_{max}$–albedo relationship[52], it is principally possible that the polarization degree of an object could be enhanced if the irregularity of its surface particles substantially raises porosity. Phaethon's surface porosity has not been directly

measured, and is not yet well constrained. Hence, we cannot rule out the possibility that larger surface porosity of Phaethon contributes to its large $P_{max}$.

Note that the above-mentioned potential causes of Phaethon's strong polarization (lower geometric albedo, prevalence of larger grains, and large surface porosity) are not mutually exclusive, and some of their combinations can be effective. Whichever of these processes (or their combinations) is causing the strong polarization of this asteroid, various ways of investigating this would serve as an important means of characterizing physical properties of Phaethon and the small solar system bodies in this category. To disentangle the combined physical processes and surface properties that involve both texture and chemical composition of small bodies of this kind, we need polarimetric observations both at small and large solar phase angles together with spectroscopic observation over a wide wavelength range as a complementary tool. A recent infrared observation that revealed that Phaethon has no absorption features at 3 μm[53] can be one step. And, partly to obtain direct answers to the questions listed above, a space mission to Phaethon named DESTINY$^+$ is planned and has now been approved by JAXA, and is awaiting its launch in 2022[54,55]. The spacecraft is supposed to make a flyby of this asteroid at a distance of 500 km or less, and is expected to provide us with high spatial resolution images containing significantly detailed information about the surface state of this asteroid. The mission outcome will unveil the nature of Phaethon's enigmatic characteristics that our polarimetric observation revealed.

## Methods

**Observations.** Nayoro Observatory, which houses the Pirka telescope, is located at the middle latitude (+44°22′25.104″N, 142°28′58.008″E, 161 m above sea level). Fortunately, several conditions, including the observatory's latitudinal location, helped us overcome the aforementioned technical difficulties of polarimetric observation of our target at large α: the observatory's location at a relatively high latitude, the season of observation (from late summer to autumn) when the Earth's North Pole is still inclined to the Sun, the location and direction of Phaethon's motion that was high above the Earth's northern hemisphere at that time, and the telescope capable of functioning safely even at very low elevation angles down to 5° without any obstacles along the line of sight. These conditions made the observation of this asteroid at large solar phase angle possible with sufficient signal-to-noise ratio ($\gtrsim$100). Table 1 shows the details of our observation.

We used the Multi-Spectral Imager (MSI)[56] installed at the $f$/12 Cassegrain focus of the Pirka telescope. MSI comprises several polarimetric devices (Wollaston prism, half-wave plate, and polarization mask), and it produces polarimetric images that cover a 3.3′ × 0.7′ field of view. We employed the standard Johnson–Cousins $R_C$-band filter for this study. This is mainly because the measurement accuracy is better in the $R_C$-band than in the $V$-band, particularly at large airmass. Individual exposure time is set to 60–180 seconds for each image, depending on the weather conditions and apparent magnitude of the asteroid. After each exposure, we routinely rotate the half-wave plate in sequence from 0° to 45.0°, from 45.0° to 22.5°, and from 22.5° to 67.5° to complete a set of polarimetric data.

**Data reduction.** We analyzed the raw data in a standard manner of astronomical image processing: All object frames are bias-subtracted and flat-fielded. Cosmic rays are removed using the L.A. Cosmic tool[57]. We extracted individual source fluxes from ordinary and extraordinary images using the aperture photometry technique implemented in IRAF. We set the aperture size for the photometry 2.5 times the full-width at half-maximum (FWHM).

To derive the polarization degree and its position angle of objects, we followed the technique implemented in ref. [31]. Specifically, we applied the following corrections: Correction of polarization efficiency, that of instrumental polarization, and that of instrumental offset in the position angle. In what follows we use the notation $q'_{pol}$ and $u'_{pol}$ for the normalized Stokes parameters instead of the conventional notation[24] $Q$, $U$, and $I$ (see Eq. (4)). We derive these parameters using the ordinary part ($\mathscr{I}_o$) and the extraordinary part ($\mathscr{I}_e$) in the extracted (observed) fluxes on the images obtained at the half-wave plate angle Ψ. More specifically, we first define quantities $R_q$ and $R_u$ as follows:

$$R_q = \sqrt{\frac{\mathscr{I}_e(0)}{\mathscr{I}_o(0)} \Big/ \frac{\mathscr{I}_e(45°)}{\mathscr{I}_o(45°)}}, \quad R_u = \sqrt{\frac{\mathscr{I}_e(22.5°)}{\mathscr{I}_o(22.5°)} \Big/ \frac{\mathscr{I}_e(67.5°)}{\mathscr{I}_o(67.5°)}}. \quad (3)$$

Then, the normalized Stokes parameters $q'_{pol}$ and $u'_{pol}$ are defined as:

$$q'_{pol} \equiv \frac{Q}{I} = \frac{1}{p_{eff}} \frac{R_q - 1}{R_q + 1}, \quad u'_{pol} \equiv \frac{U}{I} = \frac{1}{p_{eff}} \frac{R_u - 1}{R_u + 1}, \quad (4)$$

where $p_{eff}$ denotes the polarization efficiency of the total instrument system. We examined $p_{eff}$ on 1 October 2016 (during our observation runs of Phaethon) by taking dome flat images through a pinhole, and then through a Polaroid-like linear polarizer. This combination produces artificial stars with $P = 99.98 \pm 0.01\%$ in the $R_C$-band. By measuring the polarization of these artificial stars, we determined $p_{eff} = 0.9948 \pm 0.0003$ in the $R_C$-band.

The accumulation of various polarimetric observations that have been conducted at Nayoro Observatory tells us that the instrumental polarization of Pirka/MSI depends on the instrument rotator angle. We quantify this effect by inspecting the components of the Stokes parameters originating in the instrumental polarization, $q_{inst}$ and $u_{inst}$. For determining their values, we carried out an observation of an unpolarized star HD 212311[58] on 1 October 2016. The resulting values in the $R_C$-band are $q_{inst} = 0.705 \pm 0.017\%$ and $u_{inst} = 0.315 \pm 0.016\%$, respectively. Also, we define $\theta_{rot1}$ as the average instrument rotator angle during the exposures with Ψ = 0° and Ψ = 45.0°, and $\theta_{rot2}$ as the average instrument rotator angle during the exposures with Ψ = 22.5° and Ψ = 67.5°. Then, the effect of instrumental polarization is corrected by the following conversion from ($q'_{pol}$, $u'_{pol}$) to ($q''_{pol}$, $u''_{pol}$):

$$\begin{pmatrix} q''_{pol} \\ u''_{pol} \end{pmatrix} = \begin{pmatrix} q'_{pol} \\ u'_{pol} \end{pmatrix} - \begin{pmatrix} \cos 2\theta_{rot1} & -\sin 2\theta_{rot1} \\ \sin 2\theta_{rot2} & \cos 2\theta_{rot2} \end{pmatrix} \begin{pmatrix} q_{inst} \\ u_{inst} \end{pmatrix}. \quad (5)$$

Next, we correct instrumental offset in the position angle. For this purpose, we first determine the instrumental position angle offset $\theta_{off}$ through an observation of three strongly polarized stars whose position angles are well known[58] (HD 204827, HD 154445, and HD 155197). The observation was carried out on 1 October 2016 in the $R_C$-band, and it yields $\theta_{off} = 3.94 \pm 0.31°$. Then, using a parameter $\theta_{ref}$ that specifies the position angle of the instrument (which is usually set by observers, and stored as the parameter INST-PA in the FITS header), we introduce an angle $\theta'_{off}$ as

$$\theta'_{off} = \theta_{off} - \theta_{ref}. \quad (6)$$

Now we implement the correction using the following conversion formula from ($q''_{pol}$, $u''_{pol}$) into another set of parameters ($q'''_{pol}$, $u'''_{pol}$) as

$$\begin{pmatrix} q'''_{pol} \\ u'''_{pol} \end{pmatrix} = \begin{pmatrix} \cos 2\theta'_{off} & \sin 2\theta'_{off} \\ -\sin 2\theta'_{off} & \cos 2\theta'_{off} \end{pmatrix} \begin{pmatrix} q''_{pol} \\ u''_{pol} \end{pmatrix}. \quad (7)$$

Using the corrected, normalized Stokes parameters $q'''_{pol}$ and $u'''_{pol}$, we finally obtain the linear polarization degree $P$ as

$$P = \sqrt{q'''^2_{pol} + u'''^2_{pol}}, \quad (8)$$

and the position angle of polarization $\theta_P$ as

$$\theta_P = \frac{1}{2} \tan^{-1} \frac{u'''_{pol}}{q'''_{pol}}. \quad (9)$$

The linear polarization degree of an object with respect to the scattering plane (the plane where the Sun, the object, and the observer exist together) is expressed as

$$P_r = P \cos 2\theta_r, \quad (10)$$

where $\theta_r$ is given by

$$\theta_r = \theta_P - (\phi \pm 90°). \quad (11)$$

$\phi$ is the angle that determines the direction of the scattering plane on sky, and the sign in bracket is chosen to guarantee $0 \leq \phi \pm 90° < 180°$[59].

It is not that we obtain the above correction parameters ($p_{eff}$, $q_{inst}$, $u_{inst}$, $\theta_{off}$) every night. However, let us emphasize that their values did not significantly change over the two-month period of our observation. We also confirmed that the difference of their values between May 2015[31] and September–November 2016 (when we carried this work out) is smaller than the measurement error of these parameters. The reason for this parameter stability is probably that we permanently installed the imager instrument (MSI) on the telescope, and that we have not added any modifications to the instrument at all since the installation.

**Table 1 The journal of our observations**

| Median UT | $N$ | $n$ | $T_{tot}$ | $r_h$ | $\Delta$ | $\alpha$ | $P$ | $P_{err}$ | $P_r$ | $P_{r,err}$ | $\theta_r$ | $\theta_{r,err}$ | $\theta_P$ | $\theta_{P,err}$ | Airmass |
|---|---|---|---|---|---|---|---|---|---|---|---|---|---|---|---|
| 2016-09-15.476 | 16 | 4 | 32.00 | 0.780 | 0.451 | 106.5 | 50.0 | 1.1 | 49.8 | 1.1 | −2.6 | 0.6 | 77.0 | 0.6 | 4.888–6.336 |
| 2016-09-17.708 | 28 | 7 | 72.00 | 0.824 | 0.436 | 101.3 | 48.1 | 0.7 | 48.0 | 0.7 | −1.4 | 0.4 | 77.8 | 0.4 | 5.040–2.250 |
| 2016-09-24.619 | 14 | 3 | 22.75 | 0.950 | 0.407 | 85.4 | 35.6 | 0.4 | 35.6 | 0.4 | −2.7 | 0.3 | 79.2 | 0.3 | 2.438–2.240 |
| 2016-10-04.675 | 24 | 6 | 36.00 | 1.114 | 0.411 | 63.4 | 20.4 | 0.2 | 20.4 | 0.2 | −1.9 | 0.2 | 1.0 | 0.2 | 1.491–1.541 |
| 2016-10-07.636 | 48 | 12 | 58.00 | 1.159 | 0.423 | 57.7 | 15.9 | 0.1 | 15.8 | 0.1 | −2.6 | 0.2 | 24.2 | 0.2 | 1.216–1.735 |
| 2016-11-07.485 | 23 | 5 | 46.00 | 1.549 | 0.779 | 33.0 | 3.5 | 0.6 | 3.5 | 0.6 | 2.5 | 5.3 | 4.5 | 5.3 | 1.013–1.200 |

$N$: total number of exposures, $n$: total number of exposure sets, $T_{tot}$: total exposure time (minutes), $r_h$: heliocentric distance of Phaethon (au), $\Delta$: geocentric distance of Phaethon (au), $\alpha$: solar phase angle (degree), $P$: polarization degree of Phaethon (%), $P_{err}$: error of $P$ (%), $P_r$: polarization degree with respect to the scattering plane (%), $P_{r,err}$: error of $P_r$ (%), $\theta_r$: position angle with respect to the scattering plane defined in Eq. (11) (degree), $\theta_{r,err}$: error of $\theta_r$ (degree), $\theta_P$: position angle of the strongest electric vector in degrees (degree), $\theta_{P,err}$: error of $\theta_P$ (degree), airmass: range of airmass during the night

Basically $n = N/4$, but for some reason (sky situation or hardware malfunction), we have $n < N/4$ for two nights. See 00README.txt on figshare online digital repository that we mentioned in the Data availability section for more details. The airmass values are calculated from elevation angle using the airmass function implemented in IRAF. See Code availability section for more details about IRAF

**Estimate of errors**. To estimate measurement errors that $P_r$ in Eq. (10) and $\theta_r$ in Eq. (11) contain, we went through the following error estimate procedure[31]. Here we divide the errors into two classes: random errors and systematic errors.

Let us first consider random errors. We denote the normalized Stokes parameters $q'''_{pol}$ and $u'''_{pol}$ in Eq. (7) obtained from the $i$-th exposure set as $q'''_{pol,i}$ and $u'''_{pol,i}$ ($i = 1 \ldots n$). As for sources of random errors that $q'''_{pol,i}$ and $u'''_{pol,i}$ contain, we can think of shot noise from the background sky, shot noise from the asteroid itself, readout noise from the CCD, and so on. These are included in the measurement of the $4 + 4$ observed fluxes ($\mathscr{I}_e(0)$, $\mathscr{I}_o(0)$, $\mathscr{I}_e(45°)$, $\mathscr{I}_o(45°)$, $\mathscr{I}_e(22.5°)$, $\mathscr{I}_o(22.5°)$, $\mathscr{I}_e(67.5°)$, $\mathscr{I}_o(67.5°)$) appearing in Eq. (3), and are estimated by the phot function implemented in IRAF. Let us presume that each of $q'''_{pol,i}$ and $u'''_{pol,i}$ is a function of four variables: $\mathscr{I}_e(0)$, $\mathscr{I}_o(0)$, $\mathscr{I}_e(45°)$, $\mathscr{I}_o(45°)$ for $q'''_{pol,i}$, and $\mathscr{I}_e(22.5°)$, $\mathscr{I}_o(22.5°)$, $\mathscr{I}_e(67.5°)$, $\mathscr{I}_o(67.5°)$ for $u'''_{pol,i}$. We assume that no correlation exists between the errors that the four variables (fluxes) contain. Then, using the common formula for error propagation[60], we calculate variance of the random errors that are propagated to each of $q'''_{pol,i}$ and $u'''_{pol,i}$. We denote the variances as $\sigma^2_{q'''_i}$ and $\sigma^2_{u'''_i}$. Then, we derive the nightly averages of $q'''_{pol,i}$ and $u'''_{pol,i}$ through the inverse-variance weighting as follows:

$$\overline{q}'''_{pol} = \sigma^2_{\overline{q}'''_{pol}} \sum_{i=1}^n \frac{q'''_{pol,i}}{\sigma^2_{q'''_i}}, \quad \overline{u}'''_{pol} = \sigma^2_{\overline{u}'''_{pol}} \sum_{i=1}^n \frac{u'''_{pol,i}}{\sigma^2_{u'''_i}}, \quad (12)$$

where

$$\sigma^2_{\overline{q}'''_{pol}} = \frac{1}{\sum_{i=1}^n \sigma^{-2}_{q'''_i}}, \quad \sigma^2_{\overline{u}'''_{pol}} = \frac{1}{\sum_{i=1}^n \sigma^{-2}_{u'''_i}}. \quad (13)$$

The aggregated variances $\sigma^2_{\overline{q}'''_{pol}}$ and $\sigma^2_{\overline{u}'''_{pol}}$ in Eq. (13) can be regarded as synthesized random errors of $\overline{q}'''_{pol}$ and $\overline{u}'''_{pol}$ in Eq. (12).

As for systematic errors, we presume that the four parameters mentioned in Data reduction section are major contributors to these errors: polarization efficiency of the total instrument system ($p_{eff}$), instrumental polarization ($q_{inst}$ and $u_{inst}$), and instrumental position angle offset ($\theta_{off}$). Similar to the discussion when we estimate the random errors, let us regard each of $q'''_{pol,i}$ and $u'''_{pol,i}$ as a function of $p_{eff}$, $q_{inst}$, $u_{inst}$, and $\theta_{off}$. Again, let us assume that no correlation exists between the errors that the four variables contain. Using the formula for error propagation[60] again, we calculate variance of the systematic errors that are propagated to each of $q'''_{pol,i}$ and $u'''_{pol,i}$. We denote the variances as $\delta^2_{q'''_i}$ and $\delta^2_{u'''_i}$. Then, we define the eventual systematic errors of $q'''_{pol}$ and $u'''_{pol}$ for each night by making an arithmetic average of $\delta^2_{q'''_i}$ and $\delta^2_{u'''_i}$ as:

$$\delta^2_{\overline{q}'''_{pol}} = \frac{1}{n} \sum_{i=1}^n \delta^2_{q'''_i}, \quad \delta^2_{\overline{u}'''_{pol}} = \frac{1}{n} \sum_{i=1}^n \delta^2_{u'''_i}. \quad (14)$$

Adding up the random errors calculated in Eq. (13) and the systematic errors calculated in Eq. (14), the total of errors that each of $\overline{q}'''_{pol}$ and $\overline{u}'''_{pol}$ contain are expressed as follows:

$$\varepsilon_{\overline{q}'''_{pol}} = \sqrt{\sigma^2_{\overline{q}'''_{pol}} + \delta^2_{\overline{q}'''_{pol}}}, \quad \varepsilon_{\overline{u}'''_{pol}} = \sqrt{\sigma^2_{\overline{u}'''_{pol}} + \delta^2_{\overline{u}'''_{pol}}}. \quad (15)$$

Consequently, we can calculate the errors that $P$ in Eq. (8) and $\theta_P$ in Eq. (9) contain by replacing $(q'''_{pol}, u'''_{pol})$ in Eqs. (8) and (9) for $(\overline{q}'''_{pol}, \overline{u}'''_{pol})$ in Eq. (12) with each of their errors defined in Eq. (15).

During our observations at a large solar phase angle (when $\alpha > 100°$ in early September 2016), the Moon was relatively bright (the phase of the Moon was

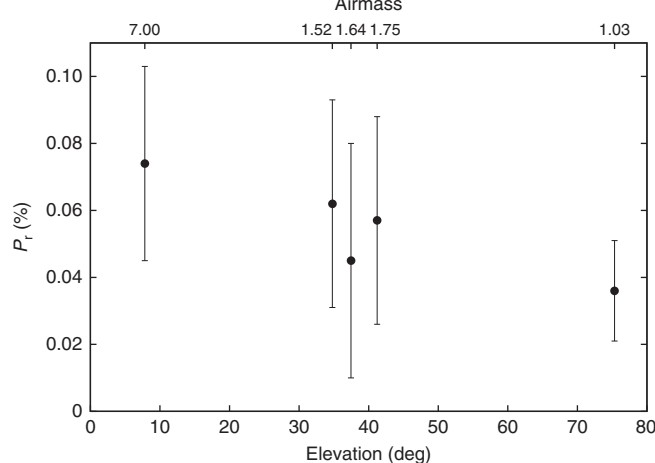

**Fig. 3** Linear polarization degree of the two unpolarized standard stars ($\theta$ UMa and HD 212311) and its dependence on airmass (elevation). We obtained the leftmost point (at airmass ~7) through the observation of $\theta$ UMa, and the other four points through the observation of HD 212311. The observation was carried out at the same observatory using the same instrument we used for our observation of Phaethon. We converted the elevation angle into airmass through the airmass function implemented in IRAF. Error bars of $P_r$ represent the sum of random errors and systematic errors that our polarimetric measurement contains. They are calculated in a manner described in Methods (see subsection Estimate of errors). Supplementary Table 3 provides actual numerical values used for this plot

approximately from 13 to 16). However, as seen in our description of the estimate process of random errors, we took shot noise from the background sky into account. This means that the influence of bright objects such as the Moon is automatically taken into consideration. Consequently, the resulting error bars appearing in Fig. 1 naturally and appropriately incorporate the influence of the Moon.

**Polarimetric dependence on airmass**. Conventionally, most astronomical observations for photometry and spectroscopy are conducted at low airmass $\lesssim 2$ (i.e., at elevation higher than about 30°). This is to eliminate the effect of atmospheric extinction. On the other hand, polarimetric analysis often ignores the airmass correction[61]. One reason is that the Earth's atmosphere has not been considered to significantly change the apparent polarimetric status of the target objects. It is also important to note that, in most cases, polarimetric observation of small solar system bodies is based on relative photometry. Intensities of the scattered light polarized along the planes perpendicular ($I_\perp$) and parallel ($I_\parallel$) to the scattering plane are simultaneously measured, and their relative value such as $\frac{I_\perp - I_\parallel}{I_\perp + I_\parallel}$ matters[23,24]. Therefore, the influence of atmospheric conditions is largely suppressed in most scenes. This is a significant difference from ordinary photometric or spectroscopic observations.

The polarimetric data we present in this paper were obtained over a wide range of solar phase angles, spanning a wide range of airmasses (from 1.013 to 6.336. see Table 1). To justify the validity and correctness of our reduction procedure applied to the data obtained at large airmass, we made additional observations of

polarimetric standard stars whose polarization degrees are well determined. Specifically, we picked two unpolarized stars ($\theta$ UMa and HD 212311) listed in ref. [58], observed them, and plotted the dependence of their $P_r$ on airmass in Fig. 3. These two standard stars are known to have very small polarization degrees ($P_r < 0.1\%$) in visible wavelengths. As we see in this figure, these stars' $P_r$ remain very small (unpolarized), regardless of the large variety of airmass used for the observation. This result justifies the validity of our measurement at large airmass, and practically guarantees the correctness of our analysis.

**Code availability**. IRAF is the Image Reduction and Analysis Facility, a general-purpose software system for the reduction and analysis of astronomical data. It is written and supported by the National Optical Astronomy Observatories (NOAO) in Tucson, Arizona, USA. NOAO is operated by the Association of Universities for Research in Astronomy (AURA) under cooperative agreement with the National Science Foundation. The code is available from http://iraf.noao.edu/.

**Data availability**. We declare that the data supporting this study's findings are available within the article and its Supplementary Information file. In addition, raw polarimetric images of the target object (Phaethon) together with bias and dome flat images that we obtained for this article are available from figshare online digital repository: https://doi.org/10.6084/m9.figshare.5873316. Raw polarimetric images of standard stars used for inspecting the dependence of $P_r$ on airmass (Fig. 3) are available from the authors upon reasonable request.

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

## Acknowledgements

The Pirka telescope is installed at Nayoro Observatory operated by the Faculty of Science at Hokkaido University, Japan. Hokkaido University takes part in the OISTER project (Optical and Infrared Synergetic Telescopes for Education and Research). We are grateful to all the Nayoro Observatory staff, particularly Yasuhiko Murakami (the observatory director), for their continuous support. We would like to thank Ryo Okazaki for providing us with data for polarimetric measurement of Phaethon at Nayoro Observatory in December 2017, as well as Tatsuharu Ono and Yuki Futamura for carrying out additional polarimetric measurement of unpolarized stars. We have also benefited from enlightening discussions with Takahiro Hiroi, Katsuhito Ohtsuka, Fumi Yoshida, and Daisuke Kuroda. T.I. is indebted to Michiru Goto who gave him administrative help to facilitate this work. Detailed and constructive editing by Yolande McLean considerably improved the presentation of the manuscript. This study is partly supported by the NRF grant 2015R1D1A1A01060025 funded by the Korean government (MEST), the JSPS Kakenhi Grant (JP25400458, JP15K13604, JP16K05546, JP18K03730), and the MEXT Kakenhi Grant (JP17H06457).

## Author contributions

T.I., M.I., and T.A. designed this study and primarily wrote the manuscript. M.I. led the observational and data analysis work with the support from M.Im., H.N., and all other authors (T.I., T.A., T.S., Y.P.B., Y.G.K., M.K., R.I.). M.W. designed and built most of the observational instrument. K.K. is responsible for the entire operation of the telescope and instrument. All authors reviewed the manuscript.

## Additional information

**Competing interests:** The authors declare no competing interests.

