## [Peer Review File · Nature Communications]

Reviewers' comments:

Reviewer #1 (Remarks to the Author):

Polarization measurements of solid bodies in the solar system have a long history, not all of it positive. The classic problem is that the interpretation of the polarization is ambiguous, because it is a function of very complicated optics in the porous regolith material that coats most solar system bodies.

Having said that, I found the Ito result to be quite unexpected and the paper to be very interesting. They have a measurement of polarization in Phaethon that is so extreme that the interpretation is relatively clear - big particles coat the surface. Even if the interpretation were suspect, Figure 2 shows that Phaethon is an extraordinary body by the standards of other solar system objects, which is itself interesting.

As an astronomer, I am interested in Phaethon because of its special role as the source of Geminids. I am not so sure of the broader interest in the community outside of astronomy: polarization is a hard-sell. This issue of the suitability for Nature, in my view, is the only weakness of the paper. If the paper is considered for Nature Astronomy, then I think it is a good match and I would support publication after re-writing in the Nature style.

Reviewer #2 (Remarks to the Author):

I read through the paper by Ito et al on the particle size distribution of grain size from polarimetry observations. While I do not have major concerns with the science presented, I am not sure if this a result that scientifically interesting or ground-breaking that warrants its publication in Nature. We have know that spectral slope and particle size are related for many years now, so what is presented in the paper is not necessarily new. I will leave it to the editor to decide if this paper should be accepted but below are some specific comments to improve the manuscript. It would also be good to repeat the observations when Phaethon makes a close flyby next month and cover the lower phase angle range if possible.

Specific Comments:

- Line 49: There are several asteroids that have blue spectra, so saying Phaethon is unique because of its blue spectrum is not accurate. Unique means one of a kind and in this case there is an entire class of asteroids that have this feature.
- Please don't use random acronyms like SSSBs. Just call them small bodies.
- Line 100: Please explain what kind of corrections you are doing to account for observations at 10 deg elevation? Also please list the airmass of the observations on Table 1.
- Line 107: Please include the observation section as appendix or supplementary material.
- The altitude of the objects when the first three measurements were made is extremely low. Typically one does not observe below 2 airmass and in the case of the first three observations it was between 3 and 5 airmass. I have serious concerns about the lack of detailed quantification of the uncertainties plotted in Figure 1.
- Figure 1: Based on Figure 1 and if we trust the error bars for the highest phase angle data points, Phaethon has similar polarimetric properties as comets 2P and 209P. Given the general acceptance that Phaethon is an active asteroid/dead comet nuclei this result should not be too surprising as the authors claim.

- Figure 2: I don't think the data on this plot are valid for the discussion. Phaethon is a dark primitive comet-type object. The data shown in this plot are all silicate-rich material and not necessarily valid for comparing with Phaethon. Ideally this should have comets and primitive asteroids rather than Icarus (which is a Q-type), Mercury, and terrestrial rocks and lunar dust.

- The general discussion towards the end of the paper is just speculative and can be trimmed down. Same goes for the last paragraph which would be obsolete by the time this paper is published in another journal.

Reviewer #3 (Remarks to the Author):

The paper presents unique measurements of the linear polarization degree of an asteroid of a rare spectral type. The observations covered a wide range of solar phase angles up to the angles above 100 deg, where the maximum of the polarization phase curve typically occurs. The authors have shown for the first time that the polarization degree of a Solar system body can reach extremely large values of 50%. It is substantially larger than the values obtained for any other small solar system bodies. This result shows that the surface properties of Solar system bodies are more diverse than considered before. The authors give important constraints on the surface characteristics of this particular asteroid Phaethon, which has been chosen as a target of future JAXA space mission.

I have two main comments to the paper:

1) there is no Table with the measured values and their errors. The values of polarization degree can be guessed from Fig.1 while the measured position angles are not given at all. The values of position angle provide an important look into possible systematic errors. In fact, the extremely large polarization was measured at a full Moon time (which is not perfect time to measure a faint object). Moreover, the instrumental polarization is rather large. The authors should provide deeper discussion on possible uncertainties of their measurements.

2) the interpretation of extremely large polarization as an indication of coarse surface grains is doubtful in my opinion. The previous single measurement of polarization of Phaethon was obtained at the phase angle of 23 deg by Fornasier et al. 2006 (the authors show this measurement in their Fig.1 but I have not found the reference to Fornasier et al. paper in the reference list). The previous measurement was close to the inversion angle and revealed that the inversion angle of Phaethon is typical for main-belt asteroids. According to laboratory measurements (the authors cited these papers) not only polarization maxima but also inversion angles depend on grain sizes. Why Phaethon has usual inversion angle and extremely large polarization maximum? It is important to discuss in the paper all possible explanations of extremely large polarization and their shortcomings.

The paper is of great interest in a wider field than planetary science providing new knowledge to our understanding of light scattering processes. The paper is worth to be published after moderate revision.

Irina Belskaya.

On the comments from Reviewer #1

#1-1 Polarization measurements of solid bodies in the solar system have a long history, not all of it positive. The classic problem is that the interpretation of the polarization is ambiguous, because it is a function of very complicated optics in the porous regolith material that coats most solar system bodies.

Having said that, I found the Ito result to be quite unexpected and the paper to be very interesting. They have a measurement of polarization in Phaethon that is so extreme that the interpretation is relatively clear - big particles coat the surface. Even if the interpretation were suspect, Figure 2 shows that Phaethon is an extraordinary body by the standards of other solar system objects, which is itself interesting.

As an astronomer, I am interested in Phaethon because of its special role as the source of Geminids. I am not so sure of the broader interest in the community outside of astronomy: polarization is a hard-sell. This issue of the suitability for Nature, in my view, is the only weakness of the paper. If the paper is considered for Nature Astronomy, then I think it is a good match and I would support publication after re-writing in the Nature style.

We appreciate very much that Reviewer #1 acknowledged the importance of our work presented in this manuscript. Through the revision that we have made this time along with the comments from the three reviewers, we believe the presentation of this manuscript is clearer for a broader community of natural sciences not only of astronomy. Also, now its format complies with the *Nature Communications* style.

In what follows in this document, we mainly argue the following two issues:

- The strong point of our study presented in this paper, as well as general complementarity of polarimetric observation to (and in some aspects, superiority over) spectroscopic observation in studies of the small solar system bodies. This is largely for responding to the comments from Reviewer #2.
- A discussion on the difference between the characteristics of P_{\max} measured at large solar phase angle and the parameters of the polarimetric negative branches ($P_{\min}, \alpha_{\text{inv}}$) measured at small solar phase angle. This is largely for responding to the comments from Reviewer #3.

We hope our argument and revisions deepen reviewers' understanding of our work presented in the manuscript.

On the comments from Reviewer #2

#2-1 I read through the paper by Ito et al on the particle size distribution of grain size from polarimetry observations. While I do not have major concerns with the science presented, I am not sure if this a result that scientifically interesting or ground-breaking that warrants its publication in *Nature*.

We are glad to know that Reviewer #2 does not have major concerns with the science that we presented in this manuscript. As we mention later in this document and also in the revised manuscript (particularly in the newly made “Methods” section), in this revision we have tried to make the presentation of the manuscript clearer and make its scientific correctness firmer, so that it better fits the standard of *Nature Communications*. Also, now the manuscript format complies with the *Nature Communications* style with the detailed “Methods” section at the trailing part of the paper. This also fulfills the requests from the reviewer (the comments #2-6, #2-7, #2-8).

#2-2 We have know that spectral slope and particle size are related for many years now, so what is presented in the paper is not necessarily new. I will leave it to the editor to decide if this paper should be accepted but below are some specific comments to improve the manuscript.

☑ **Summary of the manuscript revision:** We have added a new description as to the significance of polarimetric observation in studies of the small solar system bodies (p. 3, lines 36–42), mentioning its complementarity to spectroscopic observation.

As for the current review comment (#2-2), let us get the following points straight for clarity of discussion. Our take-away message here is that our most important accomplishment, or the key ingredient in our paper, is the result of our polarimetric observation of Phaethon as well as its implication to the surface grain size. The correlation between spectral slope and surface particle size also serves as a supporting evidence for our result, but not in the first place.

- As Reviewer #2 mentions above, the correlation between the spectral slopes of small solar system bodies and their particle (grain) size on their surface has been discussed for some years, and it is not particularly a new subject. For example, as we wrote in our manuscript, it has been experimentally confirmed that the coarse grains tend to make bluer spectrum on asteroidal (and meteoritic) surfaces.
- However, the major subject of our present paper is not spectroscopic observation for obtaining spectrum. It is about our own polarimetric observation to obtain their polarization degree. A major statement that we would like to make here is that, we accomplished a series of polarimetric of observation of an interesting small body (Phaethon), and discovered that its largest polarization degree (P_{\max}) is no less than 50%.

Here let us mention complementarity of polarimetric observation to (and in some aspects superiority over) spectroscopic observation in this line of studies of the small bodies:

- • In the spectroscopic observation of the (airless) small solar system bodies, we measure the
reflected spectrum from the solid surface of the object. But, the spectrum depends not
only on the surface texture (such as effective grain size) but also on chemical composition
of the surface material. It is generally hard for spectroscopic observation to decouple the
combined effects that can change spectroscopic slope of small bodies (particularly, objects
with featureless spectrum like Phaethon) such as chemical composition, surface grain size,
and the degree of space weathering. These combined effects can end up in complicated
changes in small body’s spectrum.
- • On the other hand, polarimetric observation of the small solar system bodies directly
measures the status of light scattering which depends on the surface texture of target
objects. This is particularly the case when we measure the maximum polarization (P_{\max})
of the object at large solar phase angle α . As for P_{\max} , famous Umow’s law has been
confirmed in its relation to albedo for various material (i.e. the higher the albedo A is,
the lower the P_{\max} value gets). And, the theoretical framework of the light scattering to
account for Umow’s law on P_{\max} based on a consideration of the frequency of multiple
light scattering is well established.
- • The frequency of multiple light scattering largely depends on albedo and grain size of
surface material of the target object. In case of Phaethon, its surface albedo has been
independently determined by mid-infrared thermal observation (e.g. Hanus̆ et al., 2016).
Therefore, it is possible to make a unique estimate of the effective grain size (through
Umow’s law) from the value of P_{\max} obtained from polarimetric observation. Validity and
accuracy of this type of estimate has been established through laboratory experiments
using terrestrial samples (e.g. Dollfus and Geake, 1977; Geake and Dollfus, 1986).
- • Based on the reasons described above, polarimetric observation to obtain P_{\max} has a
complementarity to spectroscopic observation in terms of estimating surface texture of
small solar system bodies.
- • It is needless to say that polarimetric observation has been employed not only for studies
of the small solar system bodies, but also in a wide field of astronomy and physics.
Therefore we believe that our observation result presented in the present paper will have
a ripple effect on broader field of science.

Based on the background stated above, let us describe the significance of our own study
presented in this paper again:

- • Investigation into the correlation between the maximum polarization degree of (P_{\max}) of
the actual small solar system bodies (asteroids, comets, ...) and the surface grain size
through astronomical observation does not have a great amount studies yet. Although
there have been laboratory experiments along this line using terrestrial or lunar samples
since 1970s such as what we showed in Figure 2, there are few actual observational
constraints about P_{\max} of asteroids or comets. The strongest point of our paper lies here.
- • Combining our polarimetric observational result on Phaethon with the fact that this as-
teroid greatly approaches the Sun on a regular basis, we surmise that the grain coarsening

effect due to the extensive heating (sintering) happens near at its perihelion, which grows
the grain size to a great deal.

- • In addition, very strong solar radiation pressure that works on this asteroid near its
perihelion can also be relevant, as we mentioned in the paper.
- • The large effective grain size inferred from our polarimetric observation is consistent
with the conventional knowledge known from the spectroscopic studies of this asteroid—
Phaethon’s spectrum is very blue.
- • As Umow’s law says, and as laboratory experiments have shown, solar system objects can
have large P_{\max} when their albedo is low. However, Phaethon’s albedo is not low. This
is another statement that we raise in this paper.

Making an inference on the surface grain size of small bodies through polarimetric measure-
ment has another advantage. We can think of a micro physical process that connects the strong
polarization and the coarse surface grains, even though qualitatively. As we mentioned in lines
118–127 in p. 8 of the manuscript, it is about the frequency difference of multiple scattering
of incident light at the unit optical depth. This fact (= that a picture/theory exists on micro
physical processes that account for the relation between large P_{\max} and coarse surface grains)
has an advantage over the circumstance about the correlation between the blue spectral slope
and the coarse surface grain size of small bodies: The correlation between blue spectrum and
coarse grains is still mostly empirical, and its micro physical background is not well understood.

Up to now we have stated our reasoning why we believe our polarimetric observation result is
important and useful in the context of small solar system studies in various ways. Let us also add
that Phaethon has many other interesting aspects as an active, near-Sun asteroid as well as the
parent object of a meteor stream. Therefore, we think that our detailed polarimetric observation
of Phaethon provides fundamental and underlying information on the surface environment of
small solar system bodies in a broad aspect.

Let us mention a point to note here. Different from P_{\max} , so-called the polarimetric neg-
ative branch parameters (the minimum of polarization (P_{\min}) and the solar phase angle that
marks the border between the negative branch and positive branch (α_{inv})) seem to have weaker
dependence on the surface texture of objects than P_{\max} . The negative branch parameters
seems to depend more strongly on small body’s taxonomy (e.g. Belskaya et al., 2017). It is
likely that underlying physical processes that dominate the polarimetric positive branch and
negative branch are different. But at the same time, we can say that this difference indicates
that polarimetric measurement of small bodies serves as an effective, complementary tool to
spectroscopy, particularly when it is carried out both at small and large solar phase angles.
This is because the polarimetric measurement over both negative and positive branches would
give us information on object’s surface that has gone through (at least two) different physical
processes. Please also see our responses to the review comments (#3–3 and #3–4) raised by
Reviewer #3.

**#2–3** It would also be good to repeat the observations when Phaethon makes a close flyby next
213 month and cover the lower phase angle range if possible.

As the reviewer suggested, we observed Phaethon again in 2017 December at the same
 observatory using the same instrument. As a result we obtained the polarimetric data at the
 solar phase angle between $\alpha = 33^\circ$ and 57° , and confirmed that our 2016 observation result
 is correct. The observational data at this time is still under analysis, so let us show a brief
 summary of the result:

date (UT)	α	P_r
2017-12-14	33°	$2.98 \pm 0.19 \%$
2017-12-16	57°	$14.53 \pm 0.22 \%$

We also tried to obtain polarimetric data in the range of $\alpha < 30^\circ$. But unfortunately,
 bad weather wiped out our observation opportunities, ending up having no data in this range.
 Therefore we could not obtain polarimetric data that can determine the inversion angle α_{inv} or
 the negative branch minimum P_{min} .

As for Phaethon’s polarization degree at its smaller solar phase angle, we consider that a
 past observational result (Fornasier et al., 2006) is good for comparison with ours ($\alpha = 23^\circ.0$,
 wavelength is $0.5474\mu\text{m}$, $P = 0.43\%$). Actually we had included their observational result in
 our Figure 1 (as “Phaethon ($0.5474 \mu\text{m}$)” on the second line of the legend) but we had forgotten
 to include the reference itself in the list. Reviewer #3 very kindly pointed it out. We apologize
 for causing any confusions on the review work due to our carelessness. Now we put Fornasier
 et al. (2006) in the reference list of the revised manuscript, and cited it in Figure 1’s caption.

As we reply to Reviewer #3 later in this document, Fornasier et al.’s (2006) observation
 point turns out very close to the inversion point. And, by taking a look our observational
 data and that by Fornasier et al. (2006), one finds that the fitting curve of our own $P_r(\alpha)$ can
 naturally extend toward the point that Fornasier et al. obtained. Although the observation
 wavelengths are slightly different between ours ($0.641 \mu\text{m}$) and theirs ($0.5474 \mu\text{m}$), we thus
 think these two observation are consistent to each other. This makes the coverage of solar
 phase angle of polarimetric observations of Phaethon more complete.

In the meantime, one of the authors of the present paper participated in an infrared spec-
 troscopic observation group of Phaethon during the close approach in 2017 December. We
 mention the result and its implication in our response to Reviewer #3 (#3-3) in p. 17 of this
 document, although the description is yet speculative.

**#2-4** Specific Comments:

- Line 49: There are several asteroids that have blue spectra, so saying Phaethon is unique because
 of its blue spectrum is not accurate. Unique means one of a kind and in this case there is an entire
 class of asteroids that have this feature.

Thank you very much for the comment. We found two “unique” in the Introduction section
 in the previous submission. We replaced the first one for “intriguing,” and the second one for
 “curious,” in the revised version (p. 2). We hope this rephrase protects reviewers and readers
 from getting any confusions.

**#2-5** - Please don’t use random acronyms like SSSBs. Just call them small bodies.

Thank you very much for the suggestion. Following it, in the revised manuscript we spelled
it out and just use the term “small solar system bodies” or “small bodies” as suggested.

**#2-6** - Line 100: Please explain what kind of corrections you are doing to account for observa-
tions at 10 deg elevation? Also please list the airmass of the observations on Table 1.

For the revised manuscript we made a new section named **Methods**, particularly a subsection
4.4 Polarimetric dependence on airmass in p. 17. There we put an explanation for justifying our
observational result at small elevation angles (i.e. at large airmass) which answers to reviewer’s
question. Also, for confirming the accuracy of our observational result of Phaethon at large
airmass, we observed several unpolarized stars at various airmass. The result is summarized
as Figure 3 in the **Methods** section. The bottom line is that, the Earth’s atmosphere does not
significantly change the apparent polarimetric status of the target bodies (which is measured
through relative photometry). Even when it does, the influence is very little as we see in Figure
3. Please also see our response to the review comment **#2-8**.

Along with the request from the reviewer, we also added the airmass values when we ob-
served the asteroid in Table 1 of the revised manuscript.

**#2-7** - Line 107: Please include the observation section as appendix or supplementary material.

For this purpose we made a new section **Methods**, and located it after **Discussion**. This
change also complies with the manuscript submission guideline set by *Nature Communications*.
This section describes our method of observation and analysis in detail, incorporating most of
the contents that were in the former “Observation and Data reduction” section.

**#2-8** - The altitude of the objects when the first three measurements were made is extremely
low. Typically one does not observe below 2 airmass and in the case of the first three observations
it was between 3 and 5 airmass. I have serious concerns about the lack of detailed quantification
of the uncertainties plotted in Figure 1.

Please see our response to the comment **#2-6**. Our answer to the above question here is
included there. Also, the new subsection 4.4 Polarimetric dependence on airmass (p. 17 of the
revised manuscript) gives a detailed description including a result of our addition observation
of standard “unpolarized” stars over a wide range of airmass. An important point here is that,
polarimetric observation of the small solar system bodies is based on relative photometry in
most cases. Intensities of the scattered light polarized along the planes perpendicular (I_{\perp})
and parallel (I_{\parallel}) to the scattering plane are simultaneously measured, and their relative (or
differential) value such as $P_r = \frac{I_{\perp} - I_{\parallel}}{I_{\perp} + I_{\parallel}}$ matters (e.g. Mishchenko et al., 2010; Belskaya et al.,
2015). Therefore the influence of atmospheric condition is largely suppressed in most scenes.
This is a large difference from ordinary photometric or spectroscopic observation.

**#2-9** - Figure 1: Based on Figure 1 and if we trust the error bars for the highest phase angle
data points, Phaethon has similar polarimetric properties as comets 2P and 209P. Given the general
acceptance that Phaethon is an active asteroid/dead comet nuclei this result should not be too

As is well known, Phaethon is an object that has characteristics both of an asteroid and
of a comet. Being recognized as the parent body of a major meteor stream, we may want to
say that it has a typical characteristics as a comet. As far as we are aware, however, in recent
292 years it is often emphasized that Phaethon is more like an asteroid, rather than a comet. Let
293 us itemize some studies that imply Phaethon is rather close to the asteroidal end-members:

• **Orbital origin.**

It is estimated by a series of numerical experiments as well as by spectral similarity that
Phaethon’s dynamical origin is probably in the Pallas family region, in the main asteroid
belt. It is known that the Pallas family members have blue spectrum like Phaethon, and
de León et al. (2010) found a likely dynamical path from the Pallas family region (in phase
space) to the current orbit of Phaethon, through which a Pallas family object can end
up having a Phaethon-like orbit and stay there for $O(10^7)$ years. Todorović (2018) made
the simulations more detailed, and quantitatively narrowed down the possible region that
Phaethon came through in the main belt: The 5:2 and 8:3 mean motion resonances with
Jupiter that surround the Pallas family region in the main belt. From this dynamical
viewpoint, we had better say that Phaethon has an asteroidal origin.

• **Peculiarity of the Geminid meteor stream.**

Phaethon is known to be the parent body of the Geminid meteor stream. This fact
is usually associated with a conjecture that the parent body is a comet. However in
the case of Phaethon, there is a report that the typical density of the Geminid meteors
have a much larger bulk density ($\sim 2600 \text{ kgm}^{-3}$) than cometary meteoroid (Borovička
et al., 2010). Also, significant loss of sodium has been confirmed in the the Geminid
meteor stream (e.g. Kasuga et al., 2005), indicating the influence of the solar heating has
been substantially large. These facts collectively implies that, even if we call Phaethon a
dead/extinct comet, it is very different from ordinary cometary objects.

• **Phaethon’s geometric albedo (> 0.1) is *not* low.**

If Phaethon is one of the dead/extinct comets such as 2015 TB145 (e.g. Müller et al.,
2017, whose albedo is estimated to be 5–6%) or almost an extinct one such as (3552) Don
Quixote (e.g. Mommert et al., 2014, whose albedo is estimated to be $\sim 3\%$), its albedo
would be much lower than its recently reported value of $\sim 12\%$ (e.g. Hanus et al., 2016).
The relatively high albedo that Phaethon possesses produces a remarkable difference of
this object from others, as we showed in our Figure 2. Geometric albedo of the comets
2P and 209P is as low as ~ 0.05 (see our response to the review comment #2–10).

Considering the above peculiarities and characteristics that Phaethon has, we believe this
object is not just a simple dead/extinct comet.

Let us add another point. The comets 2P and 209P are currently active, and their nuclei
have much lower albedo than Phaethon. Therefore we believe it is still surprising that Phaethon,
an asteroid or dead/extinct comet with higher albedo, exhibits similar polarimetric properties
in Figure 1 as active comets like 2P and 209P.

**#2-10** - Figure 2: I don't think the data on this plot are valid for the discussion. Phaethon is
a dark primitive comet-type object. The data shown in this plot are all silicate-rich material and
not necessarily valid for comparing with Phaethon. Ideally this should have comets and primitive
asteroids rather than Icarus (which is a Q-type), Mercury, and terrestrial rocks and lunar dust.

**Summary of the manuscript revision:** We have added data points for the
comets 2P and 209P in Figure 2 of the revised manuscript. This is for comparing the
(A, P_{\max}) values of these comets with Phaethon, and better show their differences.

As for the above comment, let us mention two points.

First, as we explained in the response to the review comment #2-9, we think that com-
parison between Phaethon and other silicate-rich material such as in our Figure 2 is valid and
meaningful. Although we admit that Phaethon sure possesses several aspects as a cometary
object, we would like to say that it is rather closer to the asteroidal end-member based on the
consideration that we stated in our response to #2-9.

Here is the second point of ours. Hinted by the above comment (#2-10), we test-revised our
Figure 2 including five more objects whose P_r data are plotted in Figure 1: 2P, 209P, Phobos,
Deimos, and Ra-Shalom. Remaking Figure 2 in this way is a crude approximation because we
do not exactly know how far/close their P_r values are from their P_{\max} except for 209 and 2P.
However, we think it is meaningful to plot their raw P_r values as the proxy of P_{\max} in order to
see the overall trend. We show the result in P2P-Figure 1.

Among the five objects, four of them (2P, 209P, Phobos, Deimos) can be categorized as
being rather primitive—two are comets, and other two are satellites with the spectrum similar
to the D-type asteroids (e.g. Rivkin et al., 2002). But as we see P2P-Figure 1, the four objects
compose a different distribution than Phaethon. Their location on this figure indicates that
their surface grains are as fine as the lunar fines or the terrestrial rock powder with the size of
$< 50\mu\text{m}$. Ra-Shalom, a C-type asteroid with a large albedo, is rather closer to Icarus. In any
case Phaethon's polarization degree is higher than any of the five objects, and it does not seem
to belong to any groups that other five objects belong to.

Based on the above discussion, we have updated Figure 2 in the revised manuscript including
the data points of for 2P and 209P whose P_r are supposed to be close to their P_{\max} .

Incidentally, we altered the description of (1566) Icarus from as being an S-type asteroid to
being a Q-type asteroid (p. 4, line 59) along with the above comment from the reviewer. We
appreciate the reviewer for pointing it out, and leading us to a more precise description.

**#2-11** - The general discussion towards the end of the paper is just speculative and can be
trimmed down. Same goes for the last paragraph which would be obsolete by the time this paper
is published in another journal.

According to this suggestion, we removed/shrunk several descriptions from this section.
First, the description about the observational opportunity in 2017 December is removed. Sec-
ond, the summaries of several lab experiments using meteorite samples about the spectral slope
and grain size was too verbose in our original manuscript, so we shrunk them a lot (now they

P2P-Figure 1: A test-revision of our manuscript’s Figure 2 including points of five more objects that are included in Figure 1: 2P, 209P, Phobos, Deimos, and Ra-Shalom. References for their albedo values are: Campins and Fernández (2002) for 2P, Kuroda et al. (2015) for 209P, Zellner and Capen (1974, their Table III) for Phobos, Thomas et al. (1996, 0.54 μm) for Deimos, and Usui et al. (2011) for Ra-Shalom. As for the comet 209P, its P_{r} value is close to its P_{\max} (Kuroda et al., 2015). The comet 2P’s P_{r} is also close to P_{\max} (Jockers et al., 2005) although the uncertainty is large. As for Ra-Shalom, we are not sure how close/far the published P_{r} value (Kiselev et al., 1999) is from its P_{\max} . As for Phobos and Deimos, their P_{r} values were measured just at a single α (Noland et al., 1973), and we cannot say their P_{r} is close to P_{\max} or not.

are confined just in a sentence: lines 130–131 in p. 9). Also, the description about the possi-
 bility that Phaethon’s surface is “dust-free” in comparison with the result of lab experiments
 has been greatly shrunk (now there is just a sentence for this: lines 157–159 in p. 10). Instead,

we added three new paragraphs in response to the requests and comments from Reviewer #3
(p. 10–11 of the revised manuscript). But we believe the scientific clarity of this section is
better now, and its presentation is more persuasive for readers.

On the comments from Reviewer #3

#3-1 The paper presents unique measurements of the linear polarization degree of an asteroid of a rare spectral type. The observations covered a wide range of solar phase angles up to the angles above 100° , where the maximum of the polarization phase curve typically occurs. The authors have shown for the first time that the polarization degree of a Solar system body can reach extremely large values of 50%. It is substantially larger than the values obtained for any other small solar system bodies. This result shows that the surface properties of Solar system bodies are more diverse than considered before. The authors give important constraints on the surface characteristics of this particular asteroid Phaethon, which has been chosen as a target of future JAXA space mission.

We thank the reviewer very much for acknowledging the novelty and importance of our work on the polarimetric observation of Phaethon. It makes our motivation on this research even bigger. Along with the comments from the reviewers, we are fully prepared to give necessary revision and improvement on the presentation of this paper.

#3-2 I have two main comments to the paper:
1) there is no Table with the measured values and their errors. The values of polarization degree can be guessed from Fig.1 while the measured position angles are not given at all. The values of position angle provide an important look into possible systematic errors. In fact, the extremely large polarization was measured at a full Moon time (which is not perfect time to measure a faint object). Moreover, the instrumental polarization is rather large. The authors should provide deeper discussion on possible uncertainties of their measurements.

As we have stated in a reply to Reviewer #2 in p. 8 of this document, we have made a new section named **Methods** in the revised manuscript. There we put a detailed explanation as to the corrections and justifications of our observational result. We also gave a description as to how we estimated the errors that the analyzed result contains, following the method applied in Ishiguro et al. (2017). As for how we deal with the background shot noise, please see the description in a paragraph in p. 17 (lines 294–299) of the revised manuscript. In a word, shot noise from the background sky including the effect of bright objects are automatically and naturally incorporated in our error analysis.

#3-3 2) the interpretation of extremely large polarization as an indication of coarse surface grains is doubtful in my opinion. The previous single measurement of polarization of Phaethon was obtained at the phase angle of 23 deg by Fornasier et al. 2006 (the authors show this measurement in their Fig.1 but I have not found the reference to Fornasier et al. paper in the reference list). The previous measurement was close to the inversion angle and revealed that the inversion angle of Phaethon is typical for main-belt asteroids. According to laboratory measurements (the authors cited these papers) not only polarization maxima but also inversion angles depend on grain sizes. Why Phaethon has usual inversion angle and extremely large polarization maximum?

Summary of the manuscript revision: We have largely modified the later part

of Discussion section in the revised manuscript (p. 10–11, lines 177–189). There, we
mentioned the points that Reviewer #3 raised, emphasizing the importance of obtaining
polarimetric characteristics both in positive and negative branches. We also made and
put a brief description of what we will discuss in what follows in this document, citing
a recent result of infrared spectroscopy of Phaethon.

We consider that the above comment raises a fundamental issue on the scientific standpoint
of our study presented in this paper. Let us appreciate Reviewer #3 for bringing up this issue,
and giving us an opportunity to self-evaluate our observational result again. In what follows
we first review that, among the polarimetric characteristics and indicators that the small solar
system bodies generally have, P_{\max} which is measured at large solar phase angle α has a deep
relationship between surface grain size of the object. Although we are quite aware that Reviewer
#3 is deeply familiar with this subject, please let us mention it for clarifying our points in this
document and in the manuscript. Next, we move on to a more specific discussion for responding
to the current review comment.

Note that it turned out that our response to the current comment (#3–3) has become very
long, largely because we think this issue is important in future studies. However, we think that
detail of what follows rather deviates from the main subject of our manuscript (presentation
of the result of polarimetric measurement of Phaethon at large solar phase angle). Also, some
of what we mention below are still speculative, and they need more intensive and quantitative
investigation. Therefore, in the manuscript we limited the description on this point brief.

Incidentally let us mention that, as we wrote in p. 7 of this document (#2–3), we had
included Fornasier et al.’s (2006) observational result in our manuscript Figure 1 (as “Phaethon
(0.5474 μm)”). But we had forgotten to include the reference itself in the previous submission,
and did not notice it until Reviewer #3 kindly pointed it out. We apologize for our carelessness.
Now we put Fornasier et al. (2006) in the reference list of the revised manuscript, and explicitly
cited it in the caption of Figure 1.

**About P_{\max}** Let us begin our discussion first with P_{\max} .

- • As for P_{\max} , Umow’s law is established and widely accepted: The smaller the albedo (A)
of an object is, the stronger its maximum polarization degree (P_{\max}) gets. Our manuscript
cites its (empirical) expression as Eq. (1) (p. 8 of the revised manuscript). Once we fix
the grain size d as a constant, Eq. (1) quantitatively yields Umow’s law.
- • As shown in Figure 2 in our paper, there have been laboratory studies on the relationship
between observed P_{\max} and surface grain size of the objects d . As yet another exposition
we have transcribed a figure from Shkuratov et al. (2017, their Figure 2) as our P2P-
Figure 2. It is a summary of laboratory experiments that indicates the relation between
the average diameter of particles d (μm) and a parameter b (which they call “polarimetric
anomaly”). Shkuratov et al. presents an experimental relation (their Eq. (2)) as $\log A +$
$a \log P_{\max} = b$ which is equivalent to Eq. (1) in our paper. As seen in P2P-Figure 2, the
grain size d increases as the parameter b increases. The increase of b leads to the increase

P2P-Figure 2: Shkuratov et al.'s (2017) Figure 2 in their p. 119. The plotted data is from Shkuratov and Opanasenko (1992).

of P_{\max} from the equation as long as other parameters (A and a) are fixed. Therefore,
 the increase of the grain size d ends up in the increase of P_{\max} .

- • As we stated in our manuscript (lines 118–127 in p. 8 of the revised version), this trend
 can be interpreted by a fact that the number of grains down to unit optical depth would
 be larger on the finer grain surface than on the coarser grain surface. The incident light
 onto the finer grain surface would experience multiple scattering more often than that
 onto the coarser grain surface. In other words, we can say that P_{\max} largely depends
 on the frequency of multiple light scattering on the material surface. This practically
 establishes Umow's law between P_{\max} and albedo A .

Here we would like to mention a point. As for the main belt asteroids (MBAs), observation
 at smaller solar phase angle α is easier to accomplish in general than observation at larger α
 due to their relative orbital geometry with respect to the Earth. Consequently, measurement
 of MBAs' P_{\min} and their inversion angle α_{inv} is more feasible than that of P_{\max} . On the other
 hand, relative orbital geometry of the near-Earth asteroids (NEAs) is very different from MBAs.
 NEAs' polarimetric observation at small α is difficult in many cases, and their P_{\min} or α_{inv} is
 not easy to obtain. However, NEAs sometimes get in a state with very large α . We grabbed one
 of the opportunities of this kind, and carried out the polarimetric measurement of Phaethon in
 2016 autumn.

Fig. 5. Relationship between P_{\min} vs. α_{inv} for asteroids of different taxonomic types. The domains for bare rocks and lunar fines as indicated by Dollfus et al. (1989) are also shown.

P2P-Figure 3: Belskaya et al.'s (2017) Figure 5. The plotted data are for the main belt asteroids.

**About P_{\min} and α_{inv}** Now, let us move on to a discussion on the negative branch of po-
 larization (represented by the quantities P_{\min} and α_{inv}) of the small solar system bodies. As
 a representative study, we have transcribed a figure from Belskaya et al. (2017, their Figure
 5) in our P2P-Figure 3. This study is about polarimetric observations of a bunch of main belt
 asteroids (MBAs) at small to medium solar phase angle, and this figure is about the relation
 between α_{inv} and P_{\min} of the sampled MBAs. In this figure, we first find that distribution
 of $(\alpha_{\text{inv}}, P_{\min})$ is strongly correlated with asteroid's taxonomy. At the same time, we see a
 dependence of the $(\alpha_{\text{inv}}, P_{\min})$ distribution on asteroids' surface texture (either bare rocks or
 fines). This distribution is based on the comparison with the laboratory experiments conducted
 in 1970s to 1980s (e.g. Geake and Dollfus, 1986). However, their dependence on the surface
 texture does not seem as strong as that on asteroid's taxonomy, as we see P2P-Figure 3. This
 behavior difference between P_{\min} (or α_{inv}) and P_{\max} can be ascribed to the difference of under-
 lying physical processes of light scattering that dominate the polarimetric negative branch and
 positive branch (e.g. Muinonen et al., 2002; Belskaya et al., 2015).

**A discussion based on a recent spectroscopic observational result** (Yet specula-
**tive, but relevant and potentially important)** As for the observational statistics presented

in P2P-Figure 3, here let us show an addition information provided from a recently obtained
observational fact about Phaethon. In summary, it turned out that Phaethon does not have
any absorption feature at the wavelength of $3\ \mu\text{m}$. This indicates that this asteroid may possess
some characteristics that are not typical of ordinary B-type asteroids.

- • As we stated in p. 7 of this document, we carried out polarimetric observation of Phaethon
again in 2017 December during its close approach to the Earth. However as we mentioned,
bad weather wiped out a large part of our observation opportunities, and we could not
complete polarimetric measurement of this asteroid that determines its P_{\min} or α_{inv} .
- • In the meantime, the third author of the present paper (Tomoko Arai) participated in an
infrared spectroscopic observation program of Phaethon during this period. The major
result is already open as an abstract that will be presented in 2018 Lunar and Planetary
Science Conference (Takir et al., 2018). This observation yielded rotationally resolved in-
frared spectra of Phaethon, and it turned out that Phaethon does not show any absorption
feature at the wavelength of $3\ \mu\text{m}$.
- • Takir et al. concludes that this result suggests that Phaethon’s surface is not hydrated,
and that Phaethon possibly had a water-rich surface in the past that became dehydrated
due to its proximity to the Sun or due to heating by impact events. We have transcribed
Takir et al.’s Figure 2 about the rotationally resolved infrared spectra of Phaethon in our
P2P-Figure 4.

Let us bring up our own opinion on Takir et al.’s result, although speculative. We think it is
possible that this outcome indicates that Phaethon, being a B-type asteroid, may also possess a
characteristics of an F-type asteroid. In the currently used SMASS asteroid classification (Bus
and Binzel, 2002) made in the wavelength between $0.44\ \mu\text{m}$ to $0.92\ \mu\text{m}$, there is no distinction
between F-type and B-type asteroids. What distinguishes F-type asteroids from the B-type
asteroids is an absence of UV absorption (e.g. Belskaya et al., 2005). But observations to
confirm this feature are not easy to carry out.

Another feature that can be unique to F-type asteroids is the non-existence of absorption
in the $3\ \mu\text{m}$ wavelength. Infrared observation in this wavelength is not easy either, but let
512 us bring up an experimental study (Hiroi et al., 1996) as an example. We transcribed their
Figure 4 in our P2P-Figure 5. Looking at P2P-Figure 5 (a), we see that the two F-type asteroids
plotted on this panel show virtually no absorption feature in $3\ \mu\text{m}$. On the other hand, a B-
type asteroid plotted here (Pallas) has the feature. Along with the comparison with thermally
metamorphosed CI/CM meteorite, the result shown in this panel also indicates that F-type
asteroids have experienced stronger heating in the past than others.

At this point, let us move back to the diagram of $(\alpha_{\text{inv}}, P_{\min})$ by Belskaya et al. (2017),
P2P-Figure 3. If Phaethon has some affinity characteristics to F-type as we conjecture, this
asteroid can be located near the F-type group in this diagram: Its inversion angle α_{inv} would
be smaller than that of typical B-type asteroids. It could also mean that this asteroid has
a surface texture rather close to “dust-free” (or, covered by coarse grains) rather than fine

P2P-Figure 4: Takir et al.'s (2018) Figure 2 that shows the rotational resolved infrared spectra of Phaethon. It was measured with the long-wavelength cross-dispersed (LXD: 1.9–4.2 μm) mode of the SpeX spectrograph/imager at the NASA Infrared Telescope Facility on 12 December 2017.

grains. This would be consistent with our conclusion from our own polarimetric observation
 of Phaethon that the effective grains on its surface are coarse, or the surface is practically free
 from fine grains (except for a short period right after its perihelion passage and temporary dust
 emission).

We thus infer that the result of our polarimetric measurement presented in this paper is
 not inconsistent to, rather in accordance with, the polarimetric measurement in the negative
 branch presented in past studies. As we have mentioned several times, we are aware that our
 statement in the current response is still speculative, and needs a lot of quantitative confir-
 mation. However, as Takir et al.'s result exemplifies, polarimetric observations both at large
 and small solar phase angles together with spectroscopic observation over wide wavelengths
 as a complementary tool are necessary for disentangling the combined physical processes and
 surface properties that involve both texture and chemical composition of the solar system small
 bodies. This is what we stated in the revised manuscript (p. 10–11).

P2P-Figure 5: Hiroi et al.'s (1996) Figure 4. The panel (a) shows the correlation between 0.7 μm band strength and the 3 μm band strength of samples. In this panel we see two "F" characters located just above the horizontal zero axis. Although there is no mention of the names of these asteroids, they are (505) Cava and (704) Interamnia according to their Table 1 (their p. 324). The "B" character located near the vertical zero axis denotes (2) Pallas. The panel (b) shows the UV absorption strength of samples against the 3 μm band strength.

#3-4 It is important to discuss in the paper all possible explanations of extremely large polarization and their shortcomings.

☑ **Summary of the manuscript revision:** We have largely modified the later part of Discussion section in the revised manuscript, particularly the two paragraphs in p. 10, lines 160–176. We mentioned two other possible causes that can enhance P_{max} . Although we still believe that the dominance of coarse grains on Phaethon's surface is the likely cause of its strong P_{max} , we added a description and mentioned a possibility that large porosity can principally increase the polarization degree of Phaethon.

In what follows we itemize three possible explanations that can cause strong polarization on the surface of the small solar system bodies, as well as their shortcomings: Low albedo, large porosity, and coarse grains.

- **Low albedo.**

When the albedo (A) of a solar system object is low, it usually shows strong polarization (P_{max}). This is nothing but Umow's law. As we mentioned in the main manuscript as

well as in this document, multiple scattering of light happens more often on the surface
with high albedo than on the surface with low albedo. As a result, the polarization degree
(particularly P_{\max}) of the surface with higher albedo gets weaker, and that with lower
albedo gets stronger. The advantage to adopt this mechanism is that, Umow’s law is well
established and theoretically explained, particularly about P_{\max} .

However, we cannot ascribe the cause of Phaethon’s strong polarization (large P_{\max}) to
its albedo—its albedo is moderately high shown in Figure 2 of our manuscript. This is
the shortcoming in this case. In addition, let us note that the dependence of P_{\min} on
albedo is not as evident as P_{\max} (Belskaya et al., 2017) in general.

• **Large porosity.**

It has been numerically confirmed that large porosity increases polarization of mate-
rial surface (e.g. Kirchschrager and Wolf, 2014; Sen et al., 2017). It is also known that
this trend holds true regardless of wavelength of incident light (e.g. Shen et al., 2009;
Kolokolova and Kimura, 2010), although its theoretical understanding is not yet com-
pletely established particularly when the wavelength is shorter than the characteristics
size of light scatterers on object surface. One of the advantages to adopt this mechanism is
that, the phenomenon has been modeled, and a number of detailed numerical simulations
have succeeded in reproducing the effect.

As we have described in the revised manuscript (p. 10, lines 171–176), we would not
completely rule out the possibility that Phaethon’s surface porosity contributes to its large
P_{\max} . In terms of geometry, surface porosity can be larger in general when the grain shape
is rough or irregular regardless of their average size. Although roughness or irregularity
of surface particles itself may not significantly affect the P_{\max} –albedo relationship (Zubko
et al., 2017), it is principally possible that the polarization degree of Phaethon could be
enhanced if the irregularity of its surface particles substantially raises the porosity.

The shortcoming of this potential cause is that, we do not have precise information on
how porous/imporous Phaethon’s (and most asteroids’) surface is. It is rather possible
that Phaethon’s surface is not quite porous, deducing from the very high bulk density of
the Geminid meteors (Borovička et al., 2010), but its direct measurement is yet to come.
Understanding the relation between Phaethon’s surface porosity and its polarimetric char-
acteristics (including both the positive and negative branches) is an important future task.
It can have an influence on broader studies of surface texture of the small solar system
bodies.

• **Coarse grains.**

As we have mentioned in this document, laboratory experiments have confirmed that the
dominance of coarse grains on material surface causes strong polarization, particularly
P_{\max} . We still believe this mechanism is the likely cause of Phaethon’s large P_{\max} . This
mechanism is consistent to other circumstances that this asteroid has, such as its very
blue spectrum, its temporary dust emission for a short period only after the perihelion
passages, and its very severe thermal environment.

As a shortcoming, we can say that the dependence of P_{\min} and α_{inv} on grain size (or on

texture) of object's surface is not as evident as that of P_{\max} . Therefore, for objects whose
P_{\max} is harder to be measured than P_{\min} (such as the main belt asteroids), additional
information should be combined with polarimetric observation.

**#3-5** The paper is of great interest in a wider field than planetary science providing new knowl-
edge to our understanding of light scattering processes. The paper is worth to be published after
moderate revision.

We again appreciate that Reviewer #3 highly evaluates the scientific value of our study. In
this revision we made as much effort as we could for improving and clarifying the presentation of
the manuscript. We hope the revised manuscript is now better ready for a broader community
of astronomical sciences.

References cited in this document

- Belskaya, I., Cellino, A., Gil-Hutton, R., Muinonen, K., and Shkuratov, Y., 2015. Asteroid po-
larimetry. In P. Michel, F. E. DeMeo, and W. F. Bottke, editors, *Asteroids IV*, pages 151–163.
The University of Arizona Press, Tucson, Arizona. [http://dx.doi.org/10.2458/azu_uapress_](http://dx.doi.org/10.2458/azu_uapress_9780816532131-ch008)
[9780816532131-ch008](http://dx.doi.org/10.2458/azu_uapress_9780816532131-ch008).
- Belskaya, I. N., Fornasier, S., Tozzi, G. P., Gil-Hutton, R., Cellino, A., Antonyuk, K., Krugly, Y. N.,
Dovgopool, A. N., and Faggi, S., 2017. Refining the asteroid taxonomy by polarimetric observations.
*Icarus*, **284**, 30–42. <http://dx.doi.org/10.1016/j.icarus.2016.11.003>.
- Belskaya, I. N., Shkuratov, Y. G., Efimov, Y. S., Shakhovskoy, N. M., Gil-Hutton, R., Cellino, A.,
Zubko, E. S., Ovcharenko, A. A., Bondarenko, S. Y., Shevchenko, V. G., Fornasier, S., and Barbieri,
C., 2005. The F-type asteroids with small inversion angles of polarization. *Icarus*, **178**, 213–221.
<http://dx.doi.org/10.1016/j.icarus.2005.04.015>.
- Borovička, J., Koten, P., Spurný, P., Čapek, D., Shrbený, L., and Šork, R., 2010. Material properties
of transition objects 3200 Phaethon and 2003 EH1. *Proceedings of the International Astronomical*
*Union*, **5** (S263), 218–222. <http://dx.doi.org/10.1017/S174392131000178X>.
- Bus, S. J. and Binzel, R. P., 2002. Phase II of the small main-belt asteroid spectroscopic survey. A
feature-based taxonomy. *Icarus*, **158**, 146–177. <http://dx.doi.org/10.1006/icar.2002.6856>.
- Campins, H. and Fernández, Y., 2002. Observational constraints on surface characteristics of comet
nuclei. *Earth Moon and Planets*, **89**, 117–134. <http://dx.doi.org/10.1023/A:1021590203207>.
- de León, J., Campins, H., Tsiganis, K., Morbidelli, A., and Licandro, J., 2010. Origin of the near-
Earth asteroid Phaethon and the Geminids meteor shower. *Astronomy and Astrophysics*, **513**, A26.
<http://dx.doi.org/10.1051/0004-6361/200913609>.
- Dollfus, A. and Geake, J. E., 1977. Polarimetric and photometric studies of lunar samples. *Philosophical*
*Transactions of the Royal Society of London Series A*, **285**, 397–402. [http://dx.doi.org/10.](http://dx.doi.org/10.1098/rsta.1977.0080)
[1098/rsta.1977.0080](http://dx.doi.org/10.1098/rsta.1977.0080).
- Fornasier, S., Belskaya, I. N., Shkuratov, Y. G., Pernechele, C., Barbieri, C., Giro, E., and
Navasardyan, H., 2006. Polarimetric survey of asteroids with the Asiago telescope. *Astronomy*
*and Astrophysics*, **455**, 371–377. <http://dx.doi.org/10.1051/0004-6361:20064836>.
- Geake, J. E. and Dollfus, A., 1986. Planetary surface texture and albedo from parameter plots of
optical polarization data. *Monthly Notices of the Royal Astronomical Society*, **218**, 75–91. [http://](http://dx.doi.org/10.1093/mnras/218.1.75)
dx.doi.org/10.1093/mnras/218.1.75.
- Hanuš, J., Delbo, M., Vokrouhlický, D., Pravec, P., Emery, J. P., Alí-Lagoa, V., Bolin, B., Devogèle,
633 M., Dyvig, R., Galád, A., Jedicke, R., Kornoš, L., Kušnirák, P., Licandro, J., Reddy, V., -P Rivet,
634 J., Világi, J., and Warner, B. D., 2016. Near-Earth asteroid (3200) Phaethon: Characterization
of its orbit, spin state, and thermophysical parameters. *Astronomy and Astrophysics*, **592**, A34.
<http://dx.doi.org/10.1051/0004-6361/201628666>.
- Hiroi, T., Zolensky, M. E., Pieters, C. M., and Lipschutz, M. E., 1996. Thermal metamorphism of
the C, G, B, and F asteroids seen from the 0.7 micron, 3 micron and UV absorption strengths
in comparison with carbonaceous chondrites. *Meteoritics and Planetary Science*, **31**, 321–327.
<http://dx.doi.org/10.1111/j.1945-5100.1996.tb02068.x>.

- Ishiguro, M., Kuroda, D., Watanabe, M., Bach, Y. P., Kim, J., Lee, M., Sekiguchi, T., Naito, H.,
Ohtsuka, K., Hanayama, H., Hasegawa, S., Usui, F., Urakawa, S., Imai, M., Sato, M., and Kuramoto,
643 K., 2017. Polarimetric study of near-Earth asteroid (1566) Icarus. *The Astronomical Journal*,
**154** (5), 180. <http://dx.doi.org/10.3847/1538-3881/aa8b1a>.
- Jockers, K., Kiselev, N., Bonev, T., Rosenbush, V., Shakhovskoy, N., Kolesnikov, S., Efimov, Y.,
Shakhovskoy, D., and Antonyuk, K., 2005. CCD imaging and aperture polarimetry of comet
2P/Encke: are there two polarimetric classes of comets? *Astronomy and Astrophysics*, **441**, 773–
782. <http://dx.doi.org/10.1051/0004-6361:20053348>.
- Kasuga, T., Watanabe, J., and Ebizuka, N., 2005. A 2004 Geminid meteor spectrum in the visible-
ultraviolet region. Extreme Na depletion? *Astronomy and Astrophysics*, **438**, L17–L20. <http://dx.doi.org/10.1051/0004-6361:200500142>.
- Kirchschrager, F. and Wolf, S., 2014. Effect of dust grain porosity on the appearance of protoplan-
etary disks. *Astronomy and Astrophysics*, **568**, A103. [http://dx.doi.org/10.1051/0004-6361/](http://dx.doi.org/10.1051/0004-6361/201323176)
[201323176](http://dx.doi.org/10.1051/0004-6361/201323176).
- Kiselev, N. N., Rosenbush, V. K., and Jockers, K., 1999. Polarimetry of asteroid 2100 Ra-Shalom:
A comparison of polarization-phase dependences for C- and S-type asteroids and comets. *Solar*
*System Research*, **33**, 192–199. <http://ads.nao.ac.jp/abs/1999SoSyR..33..192K>, translated
from *Astronomicheskii Vestnik*, **33**, 222–230, 1999.
- Kolokolova, L. and Kimura, H., 2010. Effects of electromagnetic interaction in the polarization of
light scattered by cometary and other types of cosmic dust. *Astronomy and Astrophysics*, **513**,
A40. <http://dx.doi.org/10.1051/0004-6361/200913681>.
- Kuroda, D., Ishiguro, M., Watanabe, M., Akitaya, H., Takahashi, J., Hasegawa, S., Ui, T., Kanda, Y.,
Takaki, K., Itoh, R., Moritani, Y., Imai, M., Goda, S., Takagi, Y., Morihana, K., Honda, S., Arai,
664 A., Hanayama, H., Nagayama, T., Nogami, D., Sarugaku, Y., Murata, K., Morokuma, T., Saito,
Y., Oasa, Y., Sekiguchi, K., and Watanabe, J.-i., 2015. Optical and near-infrared polarimetry for
a highly dormant comet 209P/Linear. *The Astrophysical Journal*, **814** (2), 156. [http://dx.doi.](http://dx.doi.org/10.1088/0004-637X/814/2/156)
[org/10.1088/0004-637X/814/2/156](http://dx.doi.org/10.1088/0004-637X/814/2/156).
- Mishchenko, M. I., Rosenbush, V. K., Kiselev, N. N., Lupishko, D. F., Tishkovets, V. P., Kaydash,
669 V. G., Belskaya, I. N., Efimov, Y. S., and Shakhovskoy, N. M., 2010. *Polarimetric Remote Sensing*
*of Solar System Objects*. Akadempriodyka, Kyiv, Ukraine. [http://akademperiodyka.org.ua/en/](http://akademperiodyka.org.ua/en/books/polarimetric_remote_sensing_of_Solar_System_objects)
[books/polarimetric_remote_sensing_of_Solar_System_objects](http://akademperiodyka.org.ua/en/books/polarimetric_remote_sensing_of_Solar_System_objects), full text in PDF is available
from <https://arxiv.org/abs/1010.1171>.
- Mommert, M., Hora, J. L., Harris, A. W., Reach, W. T., Emery, J. P., Thomas, C. A., Mueller,
674 M., Cruikshank, D. P., Trilling, D. E., Delbo, M., and Smith, H. A., 2014. The discovery of
675 cometary activity in near-Earth asteroid (3552) Don Quixote. *The Astrophysical Journal*, **781**, 25.
<http://dx.doi.org/10.1088/0004-637X/781/1/25>.
- Muinonen, K., Piironen, J., Shkuratov, Y. G., Ovcharenko, A., and Clark, B. E., 2002. Asteroid
photometric and polarimetric phase effects. In W. F. Bottke, Jr., A. Cellino, P. Paolicchi, and R. P.
Binzel, editors, *Asteroids III*, pages 123–138. The University of Arizona Press, Tucson, Arizona.
<http://ads.nao.ac.jp/abs/2002aste.book..123M>.
- Müller, T. G., Marciniak, A., Butkiewicz-Bąk, M., Duffard, R., Oszkiewicz, D., Käufel, H. U., Szakáts,

- R., Santana-Ros, T., Kiss, C., and Santos-Sanz, P., 2017. Large Halloween asteroid at lunar distance.
*Astronomy and Astrophysics*, **598**, A63. <http://dx.doi.org/10.1051/0004-6361/201629584>.
- Noland, M., Veverka, J., and Pollack, J. B., 1973. Mariner 9 Polarimetry of Phobos and Deimos.
*Icarus*, **20**, 490–502. [http://dx.doi.org/10.1016/0019-1035\(73\)90022-5](http://dx.doi.org/10.1016/0019-1035(73)90022-5).
- Rivkin, A. S., Brown, R. H., Trilling, D. E., Bell, J. F., and Plassmann, J. H., 2002. Near-Infrared
Spectrophotometry of Phobos and Deimos. *Icarus*, **156**, 64–75. [http://dx.doi.org/10.1006/
icar.2001.6767](http://dx.doi.org/10.1006/icar.2001.6767).
- Sen, A. K., Botet, R., Vilaplana, R., Choudhury, N. R., and Gupta, R., 2017. The effect of porosity
of dust particles on polarization and color with special reference to comets. *Journal of Quantitative
Spectroscopy and Radiative Transfer*, **198**, 164–178. [http://dx.doi.org/10.1016/j.jqsrt.2017.
05.009](http://dx.doi.org/10.1016/j.jqsrt.2017.05.009).
- Shen, Y., Draine, B. T., and Johnson, E. T., 2009. Modeling porous dust grains with ballistic
aggregates. II. Light scattering properties. *The Astrophysical Journal*, **696** (2), 2126–2137. [http://
695 //dx.doi.org/10.1088/0004-637X/696/2/2126](http://dx.doi.org/10.1088/0004-637X/696/2/2126).
- Shkuratov, I. G. and Opanasenko, N. V., 1992. Polarimetric and photometric properties of the moon:
Telescope observation and laboratory simulation. II - The positive polarization. *Icarus*, **99**, 468–484.
[http://dx.doi.org/10.1016/0019-1035\(92\)90161-Y](http://dx.doi.org/10.1016/0019-1035(92)90161-Y).
- Shkuratov, Y., Zubko, E., and Videen, G., 2017. Interpreting lunar polarimetric anomalies at large
phase angles. *Icarus*, **296**, 117–122. <http://dx.doi.org/10.1016/j.icarus.2017.05.023>.
- Takir, D., Reddy, V., Hanuš, J., Arai, T., Lauretta, D. S., Kareta, T., Howell, E. S., Emery, J. P., and
McGraw, L., 2018. 3- μ m spectroscopy of asteroid (3200) Phaethon: Implications for B-asteroids.
In *Lunar and Planetary Science Conference*, volume 2083 of *LPI Contributions*, page 2624. [https://
704 //www.hou.usra.edu/meetings/lpsc2018/pdf/2624.pdf](https://www.hou.usra.edu/meetings/lpsc2018/pdf/2624.pdf).
- Thomas, P. C., Adinolfi, D., Helfenstein, P., Simonelli, D., and Veverka, J., 1996. The surface of
Deimos: Contribution of materials and processes to its unique appearance. *Icarus*, **123**, 536–556.
<http://dx.doi.org/10.1006/icar.1996.0177>.
- Todorović, N., 2018. The dynamical connection between Phaethon and Pallas. *Monthly Notices of the
Royal Astronomical Society*, **475**, 601–604. <http://dx.doi.org/10.1093/mnras/stx3223>.
- Usui, F., Kuroda, D., Müller, T. G., Hasegawa, S., Ishiguro, M., Ootsubo, T., Ishihara, D., Kataza, H.,
Takita, S., Oyabu, S., Ueno, M., Matsuhara, H., and Onaka, T., 2011. Asteroid catalog using Akari:
AKARI/IRC mid-infrared asteroid survey. *Publications of the Astronomical Society of Japan*, **63**,
1117–1138. <http://dx.doi.org/10.1093/pasj/63.5.1117>.
- Zellner, B. H. and Capen, R. C., 1974. Photometric properties of the Martian satellites. *Icarus*, **23**,
437–444. [http://dx.doi.org/10.1016/0019-1035\(74\)90062-1](http://dx.doi.org/10.1016/0019-1035(74)90062-1).
- Zubko, E., Weinberger, A. J., Zubko, N., Shkuratov, Y., and Videen, G., 2017. Umov effect in
single-scattering dust particles: effect of irregular shape. *Optics Letters*, **42** (10), 1962–1965. [http://
718 //dx.doi.org/10.1364/OL.42.001962](http://dx.doi.org/10.1364/OL.42.001962).

**Schematic illustration of this work's framework**

For illustrating major accomplishment in our work and its relevance to Phaethon's known
characteristics, we have made a schematic figure as below. Note that this illustration is truly
schematic: The relative locations of orbits and objects are very crude, and the sizes of objects
are not at all proportional to the reality.

P2P-Figure 6: A schematic illustration of this work's framework and Phaethon.

Finally, let us sincerely thank all the three reviewers for their constructive suggestions, great
patience, and the length of time they spent for reviewing our manuscript.

Reviewers' comments:

Reviewer #3 (Remarks to the Author):

In the revised version of the manuscript the authors addressed all points raised by the reviewers. They gave very detailed replies to all comments and added new section "Methods". I have any doubts about quality of their observational results but I still have doubts about the interpretation of the measured extremely large polarization. Phaethon is an active object and it may have surface albedo variations. If the real albedo of Phaethon is lower than the value determined from radiometric measurements, the conclusion about the coarse surface of Phaethon may not be correct.

Several other comments to the text:

p.2, line 11: in Tholen's taxonomy Phaethon is classified as F-type, not B-type

p.3, line 50: "at solar phase angles larger than any other observations that have been ever reported". There are several asteroids which were observed at phase angles larger than 106 deg.

p.7, line 86: Asteroid (59) Elpis is not B-type.

p.10, line 192 "the negative branch parameters do not depend on surface texture so strongly as Pmax". It is not true. The negative branch parameters strongly depend on surface texture as shown by laboratory and numerical simulations. The fact that the negative polarization depends on asteroid's taxonomy means that surface texture are very similar for asteroids of the same taxonomic class.

On the comments from Reviewer #3

II #3-1 In the revised version of the manuscript the authors addressed all points raised by the reviewers. They gave very detailed replies to all comments and added new section “Methods”. I have any doubts about quality of their observational results

We appreciate very much that Reviewer #3 approved the quality of our polarimetric measurement and data analysis procedure.

II #3-2 but I still have doubts about the interpretation of the measured extremely large polarization. Phaethon is an active object and it may have surface albedo variations. If the real albedo of Phaethon is lower than the value determined from radiometric measurements, the conclusion about the coarse surface of Phaethon may not be correct.

☑ **Summary of the manuscript revision:** Based on a consideration on intrinsic uncertainty that Phaethon’s albedo estimate contains, we mostly adopted reviewer’s opinion (the above suggestion). Specifically, we expanded the interpretation as to what our polarimetric measurement result can mean on the surface status of this asteroid, and we proposed that the observed strong polarization implies that Phaethon’s geometric albedo is lower than the currently accepted estimate. As an alternative possibility, we placed hypotheses that relatively larger grains dominate this asteroid’s surface, and/or the asteroid’s surface porosity is large.

As a result, the following changes are made in this revision:

- Abstract → Later half is largely rewritten
- Introduction → Modified to some extent so that it fits the new discussion
- Discussion → Entirely rewritten along the review comment
- Figure 2 → Modified using two more albedo estimates of Phaethon
- Supplementary Figure 1 → Newly created and added
- Supplementary Table 2 → Extended along the revision of Figure 1
- Supplementary Table 4 → Newly created and added

As a basic premise of the following discussions, we went through all of our polarimetric measurement data again. Then, we moved on to investigating the possibility that Reviewer #3 pointed out in the later part of the above comment: “If the real albedo of Phaethon is lower than the value determined from radiometric measurements, the conclusion about the coarse surface of Phaethon may not be correct.”

Uncertainty in Phaethon’s albedo estimate

As the reviewer is quite familiar, accuracy of the estimate of Phaethon’s albedo through radiometric measurement (as well as that of many other asteroids) largely depends on how accurately its absolute magnitude (H) is determined. And, accuracy of the absolute magnitude determination depends on the accuracy of the phase curve function (i.e. the $H-G$ magnitude system) determined by photometric observation in visible wavelengths at solar phase angle α from small to large values.

Here we have to confess that, until the previous revision we had been simply convinced
that Phaethon’s geometric albedo ($p = 0.122$ presented in Hanuš et al. (2016)) based on past
radiometric observation is accurate enough. However, after receiving the review report in this
round we went through past literature again in detail, and recognized that it is not entirely
true. Phaethon’s absolute magnitude determination adopted in Hanuš et al. (2016, their p. 5)
is based on a phase curve observations whose minimum solar phase angle is no smaller than
12° . Similar situation can be seen in Ansdell et al. (2014, their Figure 2 in p. 4). This means
that, the opposition effect of this asteroid at small α has not been directly measured at all.
Ground-based observation of Phaethon at very small α is intrinsically difficult due to the relative
orbital configuration between this asteroid and the Earth. Therefore we can say that inclusion
of uncertainty into Phaethon’s absolute magnitude is inevitable. Consequently, we agree that
Phaethon’s albedo estimate can contain a non-negligible uncertainty as long as it comes from
the radiometric measurement method. We understand it is what Reviewer #3 brought up.

As an example, here we show an evaluation of how large the uncertainty of Phaethon’s albedo
can reach, depending on its absolute magnitude value. In what follows we use Phaethon’s
effective diameter ($D = 5.7$ km) recently measured through a radar observation at its close
approach to the Earth in 2017 December (Taylor et al., 2018). Measurement of size and shape
of near-Earth asteroids through active radar observation has been known reliable (e.g. Ostro,
1993; Taylor et al., 2016), and we presume the estimated effective diameter is accurate. In
Taylor et al. (2018), Phaethon’s albedo value is preliminary revised as $p = 0.10$ by assuming the
absolute magnitude $H = 14.3$ given in Hanuš et al. (2016). On the other hand, several different
estimates on Phaethon’s absolute magnitude have been published. One of the faint estimates
is presented from Minor Planet Circulars Orbit Supplement (MPO) 250234 as $H = 14.6$. If
we apply the combination of $D = 5.7$ km and $H = 14.6$ magnitude to the common formula
(e.g. Fowler and Chillemi, 1994) between asteroid’s diameter D (km), albedo p (from 0 to 1),
and absolute magnitude H

$$\log_{10} D = 0.5(6.259 - \log_{10} p - 0.4H),$$

we obtain a pretty low albedo value of $p = 0.081$. In Figure 2 of the revised manuscript, we
placed two more symbols for Phaethon using these albedo estimates (0.081 and 0.10. Note that
the albedo values have been converted into A in this figure).

As for the uncertainty of absolute magnitude determination using of the $H-G$ magnitude
system, it is known that it can reach 0.1 magnitude even for asteroids whose phase curve is
measured down to very small solar phase angle (Belskaya and Shevchenko, 2000). Recalling
the fact that the phase curve of Phaethon is measured only down to $\alpha \sim 12^\circ$, it is not hard to
imagine that its absolute magnitude contains potential uncertainties substantially larger than
0.1. This fact endorses the prospect of Phaethon’s albedo uncertainty that we roughly calculated
above—Phaethon’s albedo can be quite different from the currently accepted estimate. This
is the consideration that drove us to mostly adopt reviewer’s opinion that, Phaethon’s low
geometric albedo can be making its P_{\max} stronger.

After placing the above statement on the potentially lower albedo of Phaethon, we also
noted our conventional hypothesis that relatively larger grains may dominate this asteroid’s

surface and cause the strong polarization (which has an affinity to the fact that this asteroid
has a blue spectrum). But at the same time we are aware of, and noted that, this has a certain
inconsistency too against the polarimetric measurement results in the negative branch reported
in past literature (Fornasier et al., 2006; Belskaya et al., 2017). This is the issue that Reviewer
#3 brought up in the comments (I #3-3, I #3-4) in the first round review. Note that, together
with the above two possibilities, we also placed a description on the potentially large surface
porosity of Phaethon as the third possibility that can enhance its polarization degree.

The above listed three causes (lower albedo, prevalence of larger grains, and/or large surface
porosity) that can realize Phaethon’s strong polarization are not necessarily mutually exclusive:
some of them can be at work together to some extent. Whichever of these processes (or their
combinations) is causing the strong polarization of this asteroid, investigation on it in various
ways would serve as an important characterization of physical properties of Phaethon and the
small solar system bodies in this category. We have mentioned this point in Discussion.

Potential dependence of Phaethon’s P on its rotation phase

We checked out another possibility that Reviewer #3 raised in the first part of the review
comment II #3-2: “Phaethon is an active object and it may have surface albedo variations.” For
this purpose, we once returned to our original polarimetric measurement data (before we reduce
them into their nightly averages), and plotted its polarization degree as a function of relative
rotation phase. As for the rotation period T_{rot} we adopted the commonly accepted value of
$T_{\text{rot}} = 3.6039582$ hours (e.g. Hanuš et al., 2016). We define the time when the initial image
set on the first night was obtained (2016-09-15 11:11:06 UT) as the rotation phase = 0, and
calculated the phase of Phaethon’s rotation at the acquisition time of each image set. As
for the definition of the image “set”, please consult subsections 4.1 Observations and 4.2 Data
Reduction of the main manuscript (p. 11-13). We attached the resulting plots as Supplementary
Figure 1 together with the data table as Supplementary Table 4. We transcribed the same figure
in this document as P2P-II-Figure 1. Although individual duration in each of our observation
nights is shorter than the rotational period of this asteroid (mainly due to weather condition),
the combination of the six plots in this figure indicates that the polarimetric degree of this
asteroid does not exhibit a very remarkable dependence on its rotation phase, as long as our
observational result is concerned.

It has been recognized that Phaethon’s spectra (visible and infrared) exhibit certain varia-
tion along its rotation (e.g. Licandro et al., 2007). But its variation amplitude may be smaller
than previously considered (e.g. Takir et al., 2018). The issue of Phaethon’s possible surface
inhomogeneity, as well as the potential variation of polarimetric degree along with its rotation,
should be further investigated in future observational studies.

Avoiding the use of “coarse” / “fine” in the manuscript

Let us say that, we suspect that our use of the term “coarse” has perhaps invoked uninten-
tional confusions among reviewers. On the path of making this revision, we came into attention
that the term “coarse” is often used in various different meanings, depending on who uses it.
For example, we found that the grain size of $100 \mu\text{m}$ is regarded both as being “coarse” (e.g.
Hiroi et al., 1995; Nakamura et al., 2011) and “fine” (e.g. Yano et al., 2006) on different contexts

P2P-II-Figure 1: (Reprint of Supplementary Figure 1.) Polarization degree P of Phaethon as a function of its relative rotation phase at each observation night. The gray filled circles are the polarization degree measured at each image set. The cyan filled circles are the nightly averages of the polarization degree for each night, equivalent to what is listed in Table 1. The standard light-time correction is applied in the calculation of the rotation phase. Note that the point for set #2 of 2016-09-17 (phase = 0.58687) apparently has a lower value than others: it is perhaps possible that a nearby star's light made a small intrusion when we applied aperture photometry to the asteroid images of this set. For more detail, please consult Supplementary Figure 1 and Supplementary Table 4.

in different communities. In some cases, use of the term “coarse” automatically indicates that
 the grain size is in the order of millimeter or even larger (e.g. Gundlach and Blum, 2013). This

is different from what we intend. In order to suppress this kind of confusion, in this revision we
decided to avoid the use of the terms “coarse” and “fine” as much as possible except in com-
monly used phrases such as “grain coarsening”. Instead, we use more general terms (“large”
or “small”) for describing grain size. Also, we have tried to explicitly state that we are dealing
with the surface grain size of Phaethon that can be “larger” than the well-studied lunar and
terrestrial samples.

**II #3-3** Several other comments to the text: p.2, line 11: in Tholen’s taxonomy Phaethon is
classified as F-type, not B-type

Thank you for pointing out. It was due to our confusion with the SMASS II (Bus) classifi-
cation which categorizes Phaethon as B-type. In this revision, we corrected the sentence with
a better clarification as follows:

“This asteroid’s spectrum is categorized into B-type in the SMASS II (Bus) classi-
fication and F-type in the Tholen classification,” (p. 2, lines 10–11)

**II #3-4** p.3, line 50: “at solar phase angles larger than any other observations that have been
ever reported”. There are several asteroids which were observed at phase angles larger than 106
deg.

We are afraid that here is a little bit of misunderstanding. By the cited sentence “*In this*
*paper, we report the result of our series of polarimetric observations of Phaethon at solar phase*
*angles larger than any other observations that have been ever reported,*” we meant that our
observation is based on the widest range of solar phase angle than other polarimetric studies
of this particular asteroid (i.e. Phaethon), not of other asteroids in general. For avoiding the
confusion of this kind, in this revision we simplified the sentence as follows:

“In this paper, we report the result of our series of polarimetric observation of
Phaethon over a wide range of solar phase angle.” (p. 3, lines 48–49)

**II #3-5** p.7, line 86: Asteroid (59) Elpis is not B-type.

Thank you very much for the point. Here we meant that in the SMASS II (Bus) classification
(59) Elpis is categorized into B-type, but we did not specify which classification we used. For
avoiding misunderstanding, in this revision we placed a clear description as follows:

“. . . three asteroids ((24) Themis, (47) Aglaja, and (59) Elpis) presented in Ref 44.
They are all B-type in the SMASS II classification, as is Phaethon.” (p. 7, lines
85–87)

**II #3-6** p.10, line 192 “the negative branch parameters do not depend on surface texture so
strongly as Pmax”. It is not true. The negative branch parameters strongly depend on surface

texture as shown by laboratory and numerical simulations. The fact that the negative polarization
depends on asteroid's taxonomy means that surface texture are very similar for asteroids of the
same taxonomic class.

Thank you very much for pointing out. As we mentioned in the reply to the comment II
#3-2, we correctly understand this point now, and the description is entirely rewritten. Please
see the 2nd paragraph in p. 10 in the revised manuscript about the modified description.

Finally, let us thank again all the reviewers (including Reviewers #1 and #2) and the editors
for their great patience and the length of time that they spent for reviewing our manuscript.

References cited in this document

- Ansdell, M., Meech, K. J., Hainaut, O., Buie, M. W., Kaluna, H., Bauer, J., and Dundon, L., 2014.
Refined rotational period, pole solution, and shape model for (3200) Phaethon. *The Astrophysical*
*Journal*, **793** (1), 50. <http://dx.doi.org/10.1088/0004-637X/793/1/50>.
- Belskaya, I. N., Fornasier, S., Tozzi, G. P., Gil-Hutton, R., Cellino, A., Antonyuk, K., Krugly, Y. N.,
Dovgopol, A. N., and Faggi, S., 2017. Refining the asteroid taxonomy by polarimetric observations.
*Icarus*, **284**, 30–42. <http://dx.doi.org/10.1016/j.icarus.2016.11.003>.
- Belskaya, I. N. and Shevchenko, V. G., 2000. Opposition effect of asteroids. *Icarus*, **147**, 94–105.
<http://dx.doi.org/10.1006/icar.2000.6410>.
- Fornasier, S., Belskaya, I. N., Shkuratov, Y. G., Pernechele, C., Barbieri, C., Giro, E., and
Navasardyan, H., 2006. Polarimetric survey of asteroids with the Asiago telescope. *Astronomy*
*and Astrophysics*, **455**, 371–377. <http://dx.doi.org/10.1051/0004-6361:20064836>.
- Fowler, J. W. and Chillemi, J. R., 1994. IRAS asteroid data processing. In *The IRAS Minor Planet*
*Survey*, PL-TR-92-2049, pages 17–43. Phillips Laboratory, Directorate of Geophysics, Air Force
Material Command, Hanscom Air Force Base, MA 01731-3010, USA. [https://ntrs.nasa.gov/](https://ntrs.nasa.gov/archive/nasa/casi.ntrs.nasa.gov/19940005152.pdf)
[archive/nasa/casi.ntrs.nasa.gov/19940005152.pdf](https://ntrs.nasa.gov/archive/nasa/casi.ntrs.nasa.gov/19940005152.pdf).
- Gundlach, B. and Blum, J., 2013. A new method to determine the grain size of planetary regolith.
*Icarus*, **223**, 479–492. <http://dx.doi.org/10.1016/j.icarus.2012.11.039>.
- Hanuš, J., Delbo', M., Vokrouhlický, D., Pravec, P., Emery, J. P., Alí-Lagoa, V., Bolin, B., Devogèle,
208 M., Dyvig, R., Galád, A., Jedicke, R., Kornoš, L., Kušnirák, P., Licandro, J., Reddy, V., -P Rivet,
209 J., Világi, J., and Warner, B. D., 2016. Near-Earth asteroid (3200) Phaethon: Characterization
of its orbit, spin state, and thermophysical parameters. *Astronomy and Astrophysics*, **592**, A34.
<http://dx.doi.org/10.1051/0004-6361/201628666>.
- Hiroi, T., Binzel, R. P., Sunshine, J. M., Pieters, C. M., and Takeda, H., 1995. Grain sizes and
mineral compositions of surface regoliths of Vesta-like asteroids. *Icarus*, **115**, 374–386. [http://](http://dx.doi.org/10.1006/icar.1995.1105)
dx.doi.org/10.1006/icar.1995.1105.
- Licandro, J., Campins, H., Mothé-Diniz, T., Pinilla-Alonso, N., and de León, J., 2007. The nature
of comet-asteroid transition object (3200) Phaethon. *Astronomy and Astrophysics*, **461**, 751–757.
<http://dx.doi.org/10.1051/0004-6361:20065833>.

Minor Planet Center, 2013 January 27. Minor Planet Circulars Orbit Supplement. 250234. https://www.minorplanetcenter.net/iau/ECS/MPCArchive/2013/MPO_20130127.pdf, p. 1276.

Nakamura, T., Noguchi, T., Tanaka, M., Zolensky, M. E., Kimura, M., Tsuchiyama, A., Nakato, A.,
Ogami, T., Ishida, H., Uesugi, M., Yada, T., Shirai, K., Fujimura, A., Okazaki, R., Sandford, S. A.,
Ishibashi, Y., Abe, M., Okada, T., Ueno, M., Mukai, T., Yoshikawa, M., and Kawaguchi, J., 2011.
Itokawa dust particles: A direct link between S-type asteroids and ordinary chondrites. *Science*,
**333**, 1113–1116. <http://dx.doi.org/10.1126/science.1207758>.

Ostro, S. J., 1993. Planetary radar astronomy. *Reviews of Modern Physics*, **65**, 1235–1279. <http://dx.doi.org/10.1103/RevModPhys.65.1235>.

Takir, D., Reddy, V., Hanuš, J., Arai, T., Lauretta, D. S., Kareta, T., Howell, E. S., Emery, J. P., and
McGraw, L., 2018. 3- μm spectroscopy of asteroid (3200) Phaethon: Implications for B-asteroids.
In *Lunar and Planetary Science Conference*, volume 2083 of *LPI Contributions*, page 2624. <https://www.hou.usra.edu/meetings/lpsc2018/pdf/2624.pdf>.

Taylor, P. A., Marshall, S. E., Venditti, F., Virkki, A. K., Benner, L. A. M., Brozovic, M., Naidu,
S. P., Howell, E. S., Kareta, T. R., Reddy, V., Takir, D., Rivkin, A. S., Zambrano-Marin,
233 L. F., Bhiravarasu, S. S., Rivera-Valentn, E. G., Aponte-Hernandez, B., Sanchez-Vahamonde,
C. R., Nolan, M. C., Giorgini, J. D., Vervack, Jr., R. J., Fernandez, Y. R., Crowell, J. L., Lau-
retta, D. S., and Arai, T., 2018. Radar and infrared observations of near-earth asteroid 3200
Phaethon. In *Lunar and Planetary Science Conference*, volume 49 of *LPI Contributions*, page
2509. <https://www.hou.usra.edu/meetings/lpsc2018/pdf/2509.pdf>, [Note] The effective di-
ameter of Phaethon ($D = 5.7$ km) and the newly estimated albedo (0.10) cited in the main text
appeared in this presentation (oral talk), but they are not specifically written in the abstract. These
values are based on a personal communication provided from P. A. Taylor to Tomoko Arai.

Taylor, P. A., Richardson, J. E., Rivera-Valentin, E. G., Rodriguez-Ford, L. A., Zambrano-Marin,
242 L. F., Nolan, M. C., Howell, E. S., Benner, L. A. M., Brozovic, M., Naidu, S. P., Jao, J. S., Lee,
C. G., Giorgini, J. D., Busch, M. W., Marshall, S. E., Margot, J. L., Greenberg, A. H., Ghigo, F. D.,
Shepard, M. K., and Schmelz, J. T., 2016. Radar observations of near-Earth asteroids from Arecibo
and Goldstone. In *Lunar and Planetary Science Conference*, volume 47 of *LPI Contributions*, page
2772. <https://www.hou.usra.edu/meetings/lpsc2016/pdf/2772.pdf>.

Yano, H., Kubota, T., Miyamoto, H., Okada, T., Scheeres, D., Takagi, Y., Yoshida, K., Abe, M.,
Abe, S., Barnouin-Jha, O., Fujiwara, A., Hasegawa, S., Hashimoto, T., Ishiguro, M., Kato, M.,
Kawaguchi, J., Mukai, T., Saito, J., Sasaki, S., and Yoshikawa, M., 2006. Touchdown of the
Hayabusa spacecraft at the Muses Sea on Itokawa. *Science*, **312**, 1350–1353. [http://dx.doi.org/](http://dx.doi.org/10.1126/science.1126164)
[10.1126/science.1126164](http://dx.doi.org/10.1126/science.1126164).

REVIEWERS' COMMENTS:

Reviewer #3 (Remarks to the Author):

The authors took into account all my comments. I do not have any other comments and recommend the paper to publication.

Irina Belskaya

On the comments from Reviewer #3

III #3-1 The authors took into account all my comments. I do not have any other comments and recommend the paper to publication.

We appreciate very much that Reviewer #3 approved the significance of our work.

Let us thank once again all the three reviewers and the responsible editor for their great patience and a large amount of time that they spent for reviewing our manuscript. Their detailed and constructive comments suggested directions that significantly improved the quality of this paper. Thank you very much!